# Two different cell-cycle processes determine the timing of cell division in *Escherichia coli*

Alexandra Colin[1†‡], Gabriele Micali[2,3†], Louis Faure[1§],
Marco Cosentino Lagomarsino[4,5*], Sven van Teeffelen[1,6*]

[1]Microbial Morphogenesis and Growth Laboratory, Institut Pasteur, Paris, France; [2]Department of Environmental Microbiology, Dübendorf, Switzerland; [3]Department of Environmental Systems Science, ETH Zürich, Zürich, Switzerland; [4]IFOM, FIRC Institute of Molecular Oncology, Milan, Italy; [5]Physics Department, University of Milan, and INFN, Milan, Italy; [6]Département de Microbiologie, Infectiologie et Immunologie, Université de Montréal, Montréal, Canada

*For correspondence:
marco.cosentino-lagomarsino@
ifom.eu (MCL);
sven.vanteeffelen@gmail.com (ST)

[†]These authors contributed
equally to this work

Present address: [‡]CEA, INRA,
CNRS, UMR5168 – LPCV,
Interdisciplinary Research
Instituteof Grenoble, Université
Grenoble-Alpes, Grenoble,
France; [§]Department of
Molecular Neurosciences, Center
for Brain Research, Medical
University Vienna, Vienna, Austria

Competing interests: The
authors declare that no
competing interests exist.

Reviewing editor: Agnese
Seminara, University of Genoa,
Italy

**Abstract** Cells must control the cell cycle to ensure that key processes are brought to completion. In *Escherichia coli*, it is controversial whether cell division is tied to chromosome replication or to a replication-independent inter-division process. A recent model suggests instead that *both* processes may limit cell division with comparable odds in single cells. Here, we tested this possibility experimentally by monitoring single-cell division and replication over multiple generations at slow growth. We then perturbed cell width, causing an increase of the time between replication termination and division. As a consequence, replication became decreasingly limiting for cell division, while correlations between birth and division and between subsequent replication-initiation events were maintained. Our experiments support the hypothesis that both chromosome replication and a replication-independent inter-division process can limit cell division: the two processes have balanced contributions in non-perturbed cells, while our width perturbations increase the odds of the replication-independent process being limiting.

## Introduction

Temporal regulation of cell division is essential for cellular proliferation in all organisms. Timing of cell division determines average cell size in a population of growing cells and guarantees that every daughter cell receives one complete copy of chromosomal DNA. Despite its importance, the process remains not understood even in the best-studied model system *Escherichia coli*.

Three conceptually different classes of models have been proposed to explain division control in *E. coli* (**Figure 1B and C**).

According to the first class of models, DNA replication and segregation are regarded as limiting for cell division, while division has no influence on replication. At the single-cell level, different couplings between DNA replication and cell division have been suggested: a 'constant' (size-uncoupled) duration since the time of DNA replication initiation (C+D period in **Figure 1A**; **Ho and Amir, 2015**; **Wallden et al., 2016**), or the addition of a 'constant' (size-uncoupled) size between replication initiation and division (**Witz et al., 2019**).

A second class of models suggests that DNA replication has no direct influence on the timing of cell division under unperturbed growth conditions (**Harris and Theriot, 2016**; **Harris and Theriot, 2018**; **Si et al., 2019**; **Ojkic et al., 2019**; **Zheng et al., 2020**; **Ghusinga et al., 2016**; **Figure 1B**). Instead, a different, chromosome-independent process, the accumulation of a molecule or protein, is thought to trigger cell division, once copy number reaches a threshold. Evidence comes from the

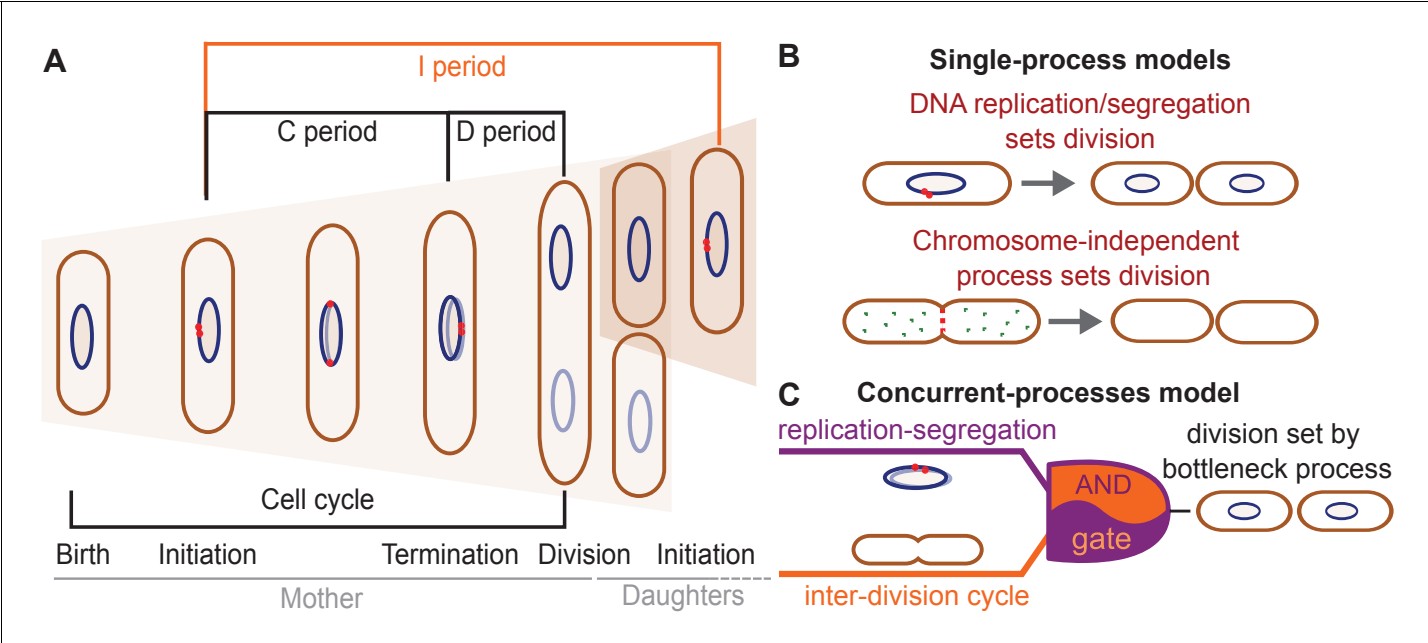

**Figure 1.** Different models have been suggested for cell-division control. (**A**) Cartoon of the cell cycle and definition of C, D and I periods. The C period is the time between initiation and termination of chromosome replication, the D period is the time between replication termination and division, and the I period is the time between subsequent initiations. (**B**) Models of cell-division control based on a single limiting process. According to the first set of models cell division is controlled by DNA replication and subsequent segregation (*Witz et al., 2019*; *Ho and Amir, 2015*; *Sompayrac and Maaloe, 1973*). According to the second set of models, cell division is controlled by a chromosome-independent inter-division process between birth and division (*Si et al., 2017*; *Si et al., 2019*; *Harris and Theriot, 2016*; *Harris and Theriot, 2018*). (**C**) Scheme of the concurrent-processes model. According to this model, the time of cell division is set by the slowest of two process, an inter-division process and chromosome replication/ segregation. When both processes are completed, the cell can go through division (analogous to an AND gate).

observation that the size added by cells between birth and division is independent of their size at birth (*Campos et al., 2014*; *Taheri-Araghi et al., 2015*; *Amir, 2014*). Further evidence comes from experiments that demonstrate the independence of this 'adder' behavior from perturbations of DNA replication (*Si et al., 2019*). Different 'accumulator' molecules have been suggested – notably cell-wall precursor molecules (*Harris and Theriot, 2016*), components of the divisome or septum (*Zheng et al., 2020*), or, more specifically, FtsZ proteins (*Si et al., 2019*; *Ojkic et al., 2019*; *Serbanescu et al., 2020*). However, whether cells effectively measure a constant size increase, whether the adder behavior emerges through the accumulation of a single molecule, and/or whether chromosome replication/segregation have a direct influence on cell division remains controversial (*Witz et al., 2019*; *Si et al., 2019*; *Zheng et al., 2020*).

A third model developed by some of us proposes that two processes limit cell division, DNA replication/segregation and a second 'inter-division' process that relates cell size at division to cell size at birth, independently of DNA replication or segregation (*Micali et al., 2018b*; *Figure 1C*). The inter-division process could be the accumulation of a molecule produced since birth, as summarized above. According to this 'concurrent-cycles' model, the slowest process sets the timing of cell division at the single-cell level. Based on recent experimental evidence (*Si et al., 2019*; *Witz et al., 2019*), DNA-replication initiation is controlled through an adder-like process between subsequent initiation events, which could also stem from a molecule accumulating during replication events (*Ho and Amir, 2015*; *Sompayrac and Maaloe, 1973*).

Micali et al. showed that single-cycle models proposed (*Wallden et al., 2016*; *Ho and Amir, 2015*; *Harris and Theriot, 2016*) fail to explain experimental data on the B and C+D subperiods in single cells, while the concurrent-cycles model is able to fit the previously available experimental datasets (*Micali et al., 2018b*). However, the model makes assumptions about the nature of the underlying processes and has more fit parameters than any of the more simple previous models. In

this situation, relevant perturbations could help us validate competing scenarios that are not simple to discern from single cells growing and dividing in standard conditions.

To test single- vs concurrent-processes models of division control, we aimed to force one of the two potentially limiting processes, the replication-independent inter-division process, to be more likely limiting for division control. *Zheng et al., 2016* showed that increasing cell width through titration of the MreB-actin cytoskeleton causes an increase of the period between replication termination and cell division (D period) without affecting the average duration of DNA replication (C period) or cell-cycle duration (see also *Si et al., 2017*). We hypothesized, that an increased D period might correspond to a decreasingly limiting role of DNA replication and an increasingly limiting role of the inter-division process for cell division.

Similar to *Zheng et al., 2016*, we thus systematically increased cell width through perturbations of the MreB actin cytoskeleton. We then followed single-cell division and DNA replication in microfluidic devices during steady-state growth conditions in minimal media, similar to previous work (*Wallden et al., 2016*; *Si et al., 2019*; *Witz et al., 2019*).

Indeed, upon increasing D period, cell size at division showed continuously decreasing correlations with cell size at initiation of DNA replication. Without any modeling, these findings already suggest that cell division is controlled by a process different from DNA replication but dependent on cell size at birth. On the contrary, in non-perturbed cells, DNA replication appears to have an important limiting role, as supported by the high correlations between division size and size at replication initiation also observed previously (*Witz et al., 2019*). By testing two recently proposed single-process models (*Si et al., 2019*; *Witz et al., 2019*) and the concurrent-process model from Micali et al., we found that only the concurrent-process model is able to describe the experimental data in both perturbed and unperturbed conditions.

In summary, our work suggests that cell division is controlled by at least two concurrent processes that link cell division to DNA replication and cell birth, respectively.

## Results

### Tracking DNA replication during steady-state growth in microfluidic channels

To investigate division control in the model organism *E. coli*, we measured cell division and DNA replication at the single-cell level using a modified wildtype strain (NCM3722, $\lambda::P_{127}$-*mcherry*, *dnaN::Ypet-dnaN*), which contains a cytoplasmic mCherry marker for accurate measurements of cell dimensions and a functional fluorescent-protein fusion to the beta-clamp of the DNA-replication machinery (YPet-DnaN), introduced at the native *dnaN* locus (*Reyes-Lamothe et al., 2010*). The YPet-DnaN fusion forms foci at the replication fork during DNA replication but is diffuse otherwise (*Figure 2A*; *Reyes-Lamothe et al., 2010*; *Moolman et al., 2014*). To investigate cells during exponential, steady-state growth conditions, we grew cells in microfluidic devices commonly referred to as 'mother machines' (*Figure 2A*, *Figure 2—video 1*), similar to previous experiments (*Wang et al., 2010*; *Long et al., 2013*; *Long et al., 2014*; *Si et al., 2019*; *Witz et al., 2019*). To reliably distinguish subsequent rounds of DNA replication, we grew cells in minimal medium (M9+NH4Cl+glycerol), such that subsequent replication rounds do not overlap.

We segmented single cells using the Oufti cell-segmentation tool (*Paintdakhi et al., 2016*) and constructed cell lineages using the Schnitzcells package (*Young et al., 2012*). We then used the YPet-DnaN signal to measure periods of DNA replication (*Figure 2—figure supplement 1*). In unperturbed cells, we found an average C period of $51 \pm 1$ min and a D period of $22 \pm 4$ min (*Supplementary file 1*), in agreement with previous bulk measurements (*Michelsen et al., 2003*). Since DnaN stays bound to DNA for about 5 min after replication termination (*Moolman et al., 2014*), we likely overestimate the average C period and underestimate the D period by this amount. However, this absolute change of period durations does not affect our investigations of cell-cycle regulation, which are based on the combined C+D period.

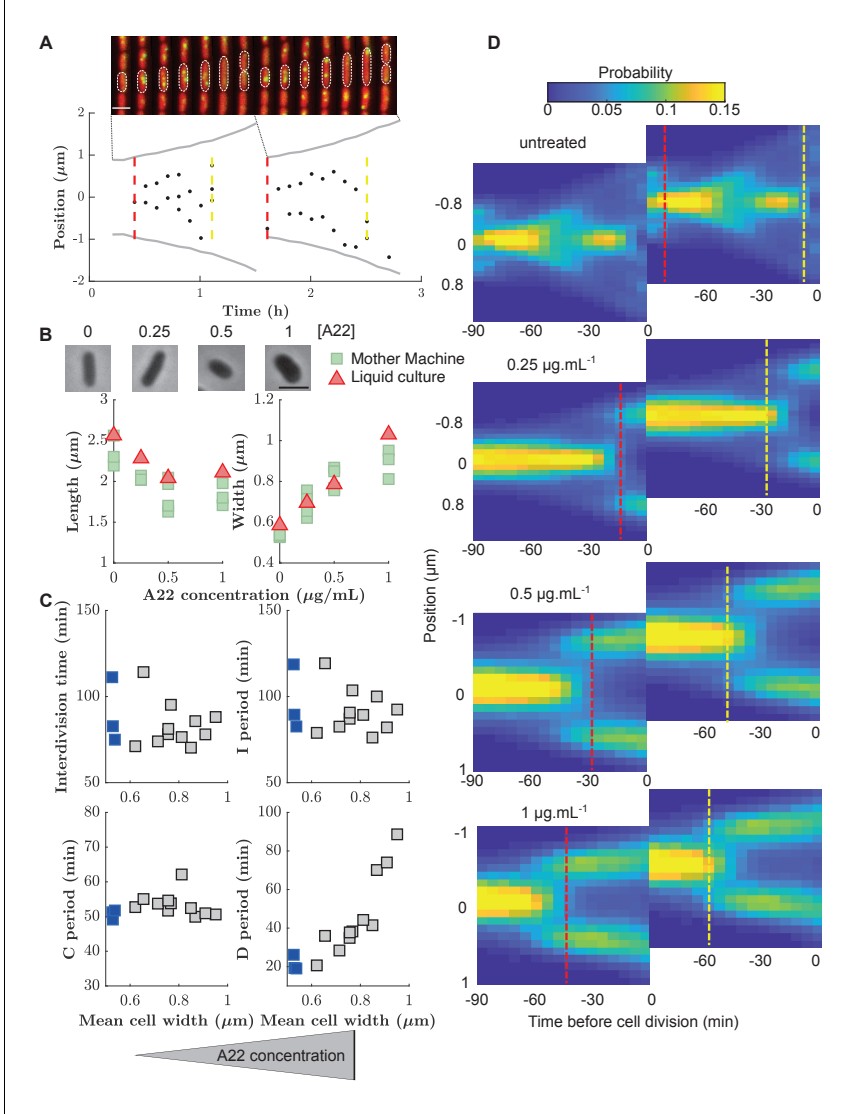

**Figure 2.** Increasing cell width through A22 increases the D period. (**A**) Top: Snapshots of a single mother-machine channel. Interval between images is 12 min. Red: cytoplasmic mCherry, yellow: YPet-DnaN. The contours show a cell growing for two consecutive cell cycles. Bottom: Cell length (gray line), the position of YPet-DnaN foci along the long axis of the cell (black dots), initiation and termination times (red and yellow dashed lines, respectively) in the same cells shown in A. Scale bar: 2 μm. (**B**) Top: Snapshots of *E. coli* S233 (NCM3722, λ::*P-mcherry, dnaN::Ypet-dnaN*) treated with sublethal amounts of A22 (concentrations in μg.mL⁻¹). Scale bar: 2 μm. Bottom: Effect of A22 treatment on average dimensions of cells grown in liquid or in mother machine for at least 6 hr of exponential growth. For cell-to-cell variations see Figure (**C**) Duration of inter-division time, I, C, and D periods as a function of average cell width measured in mother machines. Blue and gray squares represent unperturbed conditions and A22-treatment, respectively. Each symbol represents an independent biological replicate. (**D**) Conditional probability density of the occurrence of YPet-DnaN foci $p(y|t)$ as a function of cell length (y-axis) for different time points before subsequent cell division (x-axis) for different A22 concentrations as indicated on top of the maps. Maps are duplicated for better visualization of the replication process. Vertical lines indicate the beginning and end of the probability peaks that correspond to replication initiation and termination, respectively. Note that these times do not strictly agree with average replication/termination times. The online version of this article includes the following video, source data, and figure supplement(s) for figure 2:

**Source data 1.** Data used to generate *Figure 2* and its supplements.
**Figure supplement 1.** Detection of DNA replication in a single cell using the YPet-DnaN fusion.
**Figure supplement 2.** Cell-to-cell variation of cell width.
**Figure supplement 3.** Growth rate and doubling time are maintained over time and upon A22 treatment.
**Figure supplement 4.** Cell-to-cell variations of different cell-cycle periods are independent of A22 treatment.
**Figure supplement 5.** Mean initiation volume per ori as a function of average D period.
**Figure 2—video 1.** Movie of cells grown in mother machine devices for different treatments with A22.
https://elifesciences.org/articles/67495#fig2video1

## A systematic increase of cell width through the MreB-polymerization inhibitor A22 causes an increased D period

The concurrent-cycles model (*Micali et al., 2018b*) suggests that DNA replication and a replication-independent inter-division process are equally likely to limit the timing of cell division under unperturbed conditions. To test the model, and more generally the presence of two concurrent cycles, we aimed to make one of the two processes more limiting. Specifically, we speculated that the inter-division process might become the sole limiting process if the average duration between replication termination and division (D period) could be increased. Based on previous work by *Zheng et al., 2016*, we therefore systematically increased cell width by perturbing the MreB-actin cytoskeleton (*Figure 2B*). Instead of titrating MreB levels (*Zheng et al., 2016*), we treated cells with sub-inhibitory concentrations of the MreB-polymerization inhibitor A22 (*Bean et al., 2009*), similar to previous studies (*Tropini et al., 2014*).

Increasing A22 concentration leads to increasing steady-state cell width both in batch culture and in the mother machine (*Figure 2B*), without affecting cell-to-cell width fluctuations (*Figure 2—figure supplement 2*), and without affecting doubling time (*Figure 2C*) or single-cell growth rate (*Figure 2—figure supplement 3*). Furthermore, growth-rate fluctuations remain constant (*Figure 2—figure supplement 4A*) and similar to previous measurements (*Kennard et al., 2016*; *Grilli et al., 2018*).

In line with the results of *Zheng et al., 2016*, the increase of cell width leads to an increase in the average D period (*Figure 2C*) as hypothesized. At the same time, the average C period (*Figure 2C*) and the average cell volume at the time of replication initiation remain unperturbed (*Figure 2—figure supplement 5*), as previously reported (*Zheng et al., 2016*). Cell-to-cell fluctuations in the duration of sub-periods remain constant (I, C, and interdivision periods) or decrease mildly (D period) (*Figure 2—figure supplement 4B*). While sub-periods are extracted from single-cell lineages, the shift of replication to earlier times is also observed in the probability distributions of replicase positions (*Figure 2D*), where periods of both early and late replication appear as marked foci. Vertical lines that indicate the beginning or end of peaks in *Figure 2D* are guides to the eye and should not be interpreted as average times of initiation or termination.

## Increasing D period through A22 leads to decreasing correlations between DNA replication and cell division

In view of the previously suggested concurrent-cycles model (*Micali et al., 2018b*), we speculated that DNA replication might not be limiting for cell division if the D period was increased, while a replication-independent inter-division process might become the sole limiting process for cell division. Alternatively, as previously suggested (*Zheng et al., 2016*), replication could still be the limiting process determining the timing of cell division, for example through a width-dependent added size between replication initiation and subsequent cell division (*Witz et al., 2019*).

The coupling between cell size and cell growth over different cell-cycle subperiods can be quantified in different ways (*Jun and Taheri-Araghi, 2015*; *Osella et al., 2017*; *Cadart et al., 2019*). For convenience, and following *Jun and Taheri-Araghi, 2015*; *Micali et al., 2018b*; *Si et al., 2019*; *Ho and Amir, 2015*, we quantified behavior during different sub-periods using 'adder plots', which display the added size during the period *versus* the initial size, both normalized by their means (see Materials and methods for a discussion of the use of length instead of volume as a proxy for size). We refer to the slope of these plots as 'coupling constants' $\zeta_X$, where $X$ denotes the respective sub-period. A coupling constant of 0 corresponds to adder behavior. A coupling constant of 1 corresponds to a 'timer' process, that is a process that runs for a constant duration on average, independently of cell size at the beginning of the period, and a coupling constant of -1 corresponds to a process where the final size is independent of the size at the beginning of the period (see Materials and methods).

First, we measured the added size between birth and division. In agreement with previous results (*Campos et al., 2014*; *Taheri-Araghi et al., 2015*), untreated cells showed 'adder behavior', that is, the added size between birth and division is independent of birth size $L_0$, with a coupling constant (or slope) of $\zeta_G = -0.046 \pm 0.085$ (*Figure 3A*). Here, the uncertainty denotes the standard deviation between biological replicates (*Supplementary file 1*). With increasing D period duration (through increasing A22 concentration), cells continued to show near-adder behavior with a weak trend

towards sizer behavior (*Figure 3B*). For single-cell point clouds of intermediate A22 concentrations see *Figure 3—figure supplement 1*. Similarly, cells also show adder behavior between subsequent rounds of replication initiation (*Figure 3C*). More specifically, cells add a constant size per origin of replication between subsequent rounds of initiation, independently of initial initiation size ($\zeta_I = -0.013 \pm 0.098$). This behavior is robust with respect to variations of average growth rate using a poorer growth medium (*Figure 3—figure supplement 3*). For unperturbed cells, this behavior was previously proposed theoretically (*Ho and Amir, 2015*; *Sompayrac and Maaloe, 1973*) and demonstrated experimentally (*Si et al., 2019*; *Witz et al., 2019*). *Ho and Amir, 2015* previously demonstrated that the average size per origin and average added size per origin are equal to one another during steady-state growth. The scaling of average cell size at initiation with the number of replication origins initially deduced by *Donachie, 1968* and later confirmed for different growth rates (*Wallden et al., 2016*) and for different cell widths (*Zheng et al., 2020*) is therefore also a strong motivation to consider the added size per origin (rather than the non-normalized added size) in our and previous single-cell studies (*Si et al., 2019*; *Witz et al., 2019*).

We found that $\zeta_I$ is constant, independently of A22 treatment (*Figure 3D*). Together with the constancy of the average initiation volume (*Figure 2—figure supplement 5*, *Ho and Amir, 2015*; *Si et al., 2017*; *Zheng et al., 2016*) this suggests that the process of replication initiation is not affected by the A22-induced cell widening.

In contrast to the weak dependency of $\zeta_G$ and $\zeta_I$ on drug treatment, correlations between initiation size and corresponding cell division systematically change as a function of average D period (*Figure 3G–H*). While unperturbed cells effectively show adder behavior ($\zeta_{CD} = -0.10 \pm 0.11$, *Figure 3E*), in agreement with the analysis of previous experimental data (*Micali et al., 2018b*; *Witz et al., 2019*), $\zeta_{CD}$ continuously changes toward a value of -1 with increasing average D period (*Figure 3F*). Note that the negative value of $\zeta_{CD}$ corresponds to a lack of correlation between division size and size at initiation (*Figure 3G*), typically predicted by the models where replication is never limiting for cell division *Micali et al., 2018b*; *Si et al., 2019*. This lack of correlations can also be illustrated differently: Division size is decreasingly dependent of the size at initiation with increasing D period (*Figure 3H*).

With increasing average D period, replication is increasingly likely to happen in the mother cell (*Figure 2D*). To test whether this behavior might be responsible for a change of the slopes of the point clouds observed in *Figure 3E–F*, we separated the single-cell measurements of untreated cells or cells treated with a low A22 concentration (0.25 µg/ml) into separate clouds, depending on whether initiation happened in the mother or in the daughter cell, respectively (*Figure 3—figure supplement 2*). We did not observe a separation of point clouds nor differences between their slopes, suggesting that the spread of the C period over a division event does not affect correlations between initiation and division or between subsequent initiation events.

From these observations, we conclude that with increasing average D period a process different from DNA replication is likely increasingly responsible for division control.

## A replication-independent adder-like process is increasingly likely the bottleneck process for cell division

As described in the introduction, a range of different single-process models were proposed in the past to explain correlations between DNA replication and cell division (*Si et al., 2019*; *Harris and Theriot, 2016*; *Witz et al., 2019*; *Wallden et al., 2016*; *Ho and Amir, 2015*). Some of us recently argued that existing single-process models are incapable to reconcile correlations observed in previous experimental datasets (*Micali et al., 2018b*), which led us to propose the concurrent cycle scheme illustrated in *Figure 4A*. The model assumes two processes that must both finish for cell division to occur, one replication/segregation process related to the size at replication initiation and one inter-division process related to the size at birth. The model contains three control parameters: $\zeta_{CD'}$ controls the replication/segregation process and $\zeta_H$ controls the inter-division process. A third parameter, $\zeta_I$ controls the inter-initiation process that relates replication initiation to the cell size at the previous initiation. The slopes of the inter-division period ($\zeta_G$) and of the C+D period ($\zeta_{CD}$) emerge from the competition of the two cycles and are predictions of the model.

To fit the concurrent-cycles model to our experimental data, we set the inter-initiation process to be an adder ($\zeta_I = 0$), based on our experimental results (*Figure 3C*), in agreement with previous observations in unperturbed cells (*Si et al., 2019*; *Witz et al., 2019*). Furthermore, we assumed that

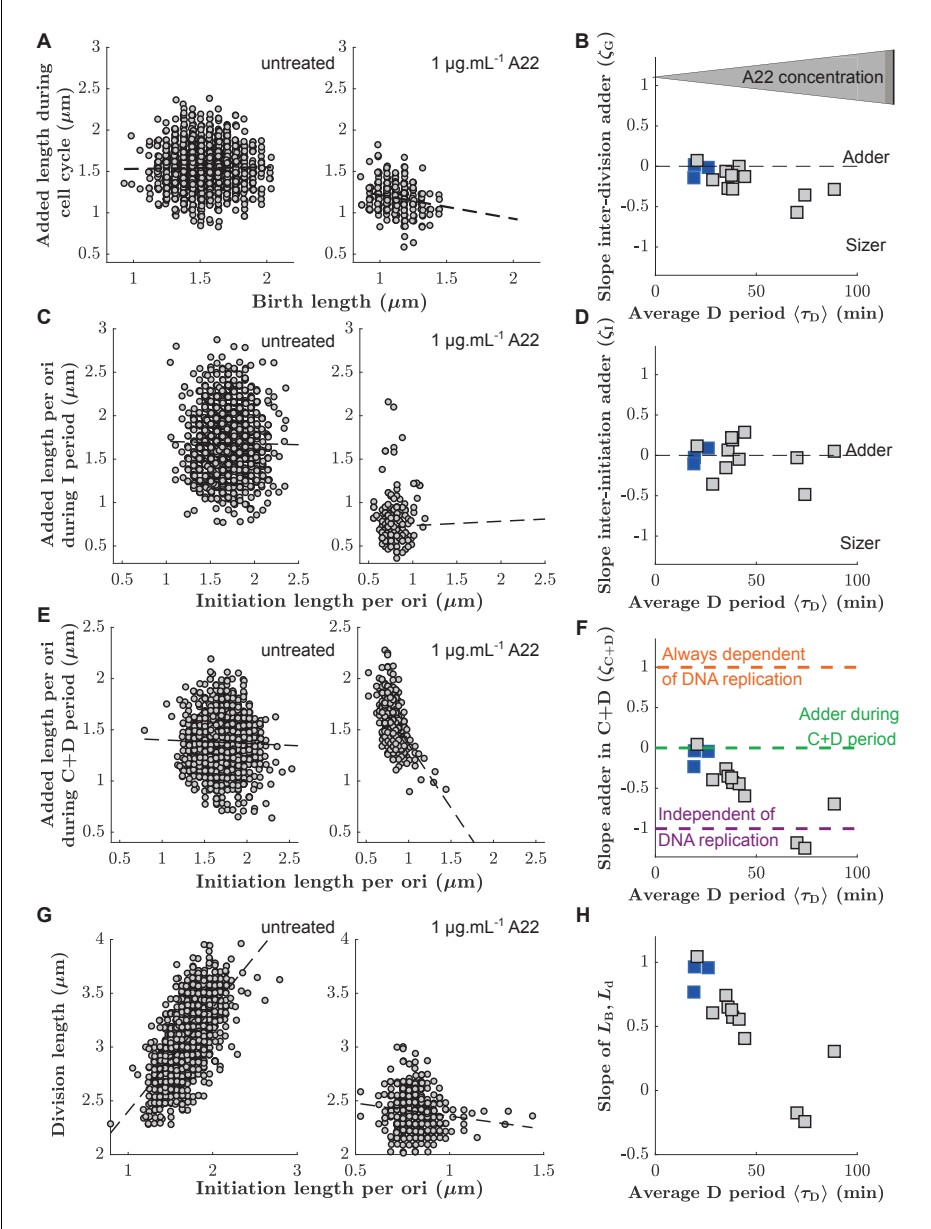

**Figure 3.** Single-cell correlations between division and DNA replication events. (A,C,E) Added size between birth and division (A), between subsequent events of replication initiation (C), and during the C+D period (E), for untreated cells (left) and cells treated with 1 µg.mL⁻¹ A22 (right). Points represent single cells. Dashed lines represent robust linear fits. All lengths are indicated in units of µm. (B,D,F) Slopes of the added sizes corresponding to A, C, E, respectively, as a function of the D period as obtained through sub-lethal A22 treatment (0–1 µg.mL⁻¹). A slope of 0 represents adder behavior, while a slope of -1 represents independence on the size at the beginning of the sub-period (sizer behavior). Blue and gray squares represent unperturbed conditions and A22-treatment, respectively. Each symbol represents an independent biological replicate. (G,H) Division size $L_d$ as a function of initiation size per ori $L_B/n_{Ori}$ (G) and corresponding slopes (H) in analogy to panels A, B, respectively. The decreasing slope in H demonstrates decreasing dependency of division on DNA replication.

The online version of this article includes the following source data and figure supplement(s) for figure 3:

**Source data 1.** Data used to generate *Figure 3* and its supplements.

**Figure supplement 1.** Added lengths during different subperiods as a function of the size at the beginning of the respective subperiod for different A22 concentrations.

**Figure supplement 2.** Added lengths during C+D period (top) or I period (bottom) as a function of the size at initiation for untreated cells (left) and cells treated with 0.25 µg/ml A22 (right).

**Figure supplement 3.** Adder behaviors for cell cycle, I period, and C+D period are robustly maintained at two different growth rates.

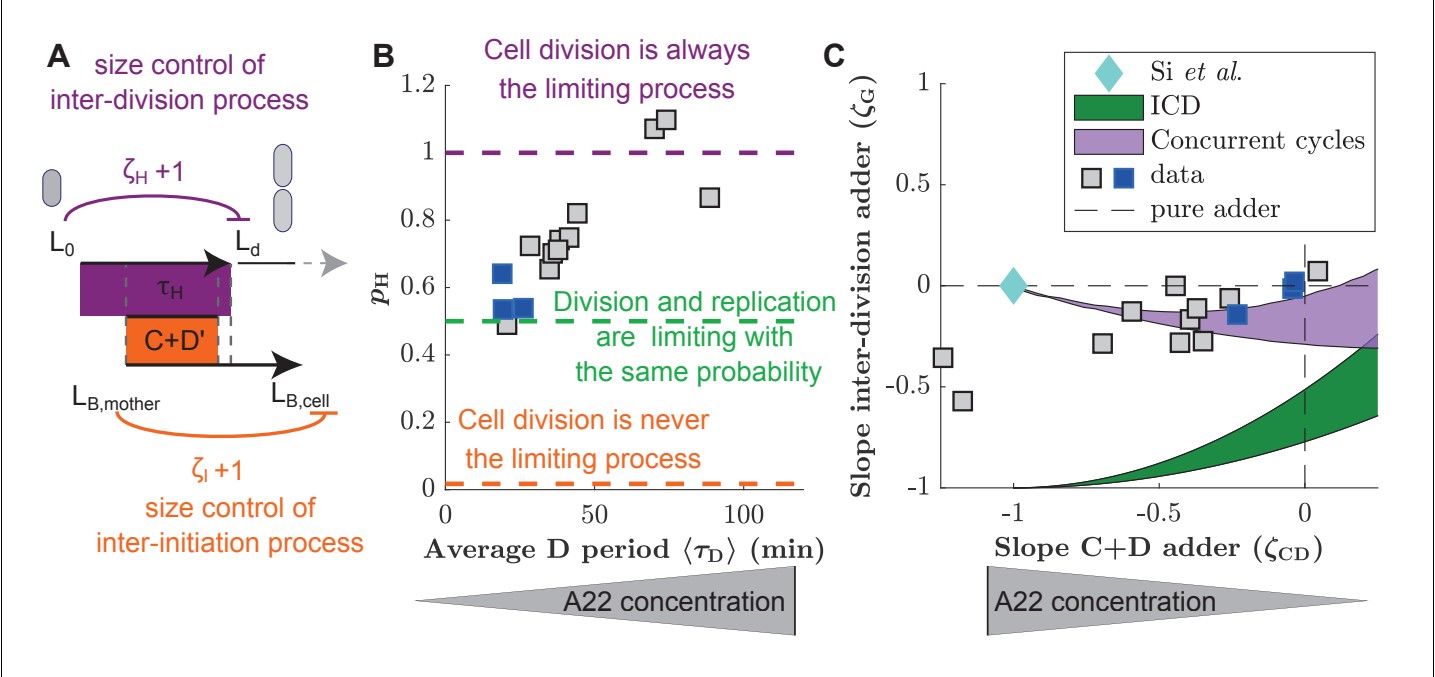

**Figure 4.** Experimental validation of the concurrent cycles model. (A) Cartoon: Two independent inter-division and timer-like replication/segregation must be completed before division occurs. The inter-division process is assumed to exhibit adder-like behavior with control parameter $\zeta_H = 0$, while the replication/segregation is a timer (see Materials and methods for details on the estimation). The adder-like inter-initiation processes with control parameter $\zeta_I = 0$ determines size at initiation. (B) Model-fitting to experimental data reveals the probability $p_H$ of the inter-division process to control cell division as a function of increasing D period (with increasing A22 concentration), assuming constant control parameters $\zeta_H = 0$ and $\zeta_I = 0$. (C) Slopes of adder plots $\zeta_G$ as a function $\zeta_{CD}$. Blue diamond: prediction in *Si et al., 2019*. Dotted lines: Prediction of pure adder models. Green: Prediction from a general class of single-process chromosome-limited models ('ICD' models, see Supplementary Notes) (*Micali et al., 2018b*), where cells divide after completion of the replication/segregation process with variable $\zeta_{CD}$. Purple: Prediction of the concurrent cycles model. Shaded areas represent the ranges of predictions using the maximum and minimum experimentally measured input parameters (ratio of variance of size at initiation over size at birth; ratio of mean size at division over size at birth). (B, C) Blue and gray squares represent unperturbed conditions and A22-treatment, respectively. Each symbol represents an independent biological replicate.

The online version of this article includes the following source data and figure supplement(s) for figure 4:

**Source data 1.** Data used to generate *Figure 4* and its supplements.
**Figure supplement 1.** Predictions of the concurrent cycles model if $\zeta_H$ is left as a free parameter.
**Figure supplement 2.** Asymmetric division drives C+D to adder behavior in ICD single-process models.
**Figure supplement 3.** Comparison of adder slopes for data of unperturbed cells generated by different labs.
**Figure supplement 4.** Theoretical predictions in the small-noise approximations agree with simulations at realistic noise levels.
**Figure supplement 5.** The consideration of asymmetry of cell division has no significant effect on slopes of inter-initiation adder and adder during C+D period.

replication segregation (the C+D' perid) is a timer process ($\zeta_{CD'} = 1$) that requires a minimum time to be completed, which is independent of size at the time of initiation, and does not vary in A22 perturbations. Note that neither the minimum completion time C+D' nor the coupling parameter $\zeta_{CD}$ can be measured experimentally, or bypassed in the model. In principle these parameters could change under A22 perturbations, since MreB affects the activity of topoisomerase IV (*Madabhushi and Marians, 2009*; *Kruse et al., 2003*), an enzyme that mediates the dimerization of sister chromosomes. However, constancy of $\zeta_{CD'}$ is supported by the constancy of the C period, and the minimum D' period cannot increase too strongly with width in the model, because otherwise it would render replication/segregation limiting for division under A22 perturbations, contrary to our experimental observation. Hence, for simplicity, we assumed $\zeta_{CD'}$ and the D' period to stay constant. For the inter-division process we assumed that $\zeta_H = 0$ (adder). This assumption is supported by previous experiments in filamentous cells, transiently inhibited for division (*Wehrens et al., 2018*). Those cells divide following a cell-cycle adder and therefore divide much more frequently than non-filamentous cells, likely because DNA replication is never limiting. The adder hypothesis is also

compatible with the accumulation models of FtsZ or other divisome/septum components for this sub-period, as recently hypothesized *Si et al., 2019*; *Zheng et al., 2020*; *Ojkic et al., 2019*.

Compared to a single-process model, this framework outputs the extra parameter $p_H$, which quantifies the probability that the inter-division process is limiting. *Figure 4B* shows how by fitting the model to our data, increasing D period duration leads to an increase of $p_H$. The model therefore predicts that the two independent processes, DNA replication and a replication-independent inter-division process, are almost equally likely to limit cell division under unperturbed conditions (*Micali et al., 2018b*). However, with increasing average D period through perturbation by A22, the replication-independent inter-division process is increasingly likely limiting for cell division.

In a generalized framework, we also allowed the inter-division control parameter $\zeta_{\mathrm{H}}$ to vary, fitting $\zeta_{\mathrm{H}}$ and $p_H$ simultaneously, at the cost of an extra parameter. We found that $\zeta_{\mathrm{H}}$ decreases mildly from an adder-like behavior toward a sizer with increasing average D period (*Figure 4—figure supplement 1B*). $p_H$ increases with the D period regardless of the fitting strategy (*Figure 4—figure supplement 1A*).

Two recent studies have proposed single-process models based on new experimental data: First, a chromosome-limited model that links replication and subsequent division through an adder process (*Witz et al., 2019*), which is the best-fitting model of a whole class of models where replication is limiting and initiation is set by an adder ('ICD' models, see Supplementary Notes) and second, a chromosome-agnostic model that considers replication and division processes as independent of one another (*Si et al., 2019*). We therefore tested the performance of both of these models on our experimental data of unperturbed cells, by jointly comparing the predicted couplings of the inter-division period and the C+D period. We found that both frameworks appear to be incompatible with our data (*Figure 4C*).

We also verified that the concurrent-cycles scenario generally shows better agreement with recently published data (*Si et al., 2019*; *Witz et al., 2019*) than single-process models (*Figure 4—figure supplement 3*). Interestingly, when fitting our model to all datasets including our own, we found that $p_H \approx 0.5$ at slow growth (if the average doubling time is smaller than 1.4 hr), while $p_H$ increases with decreasing doubling time. This trend is in qualitative agreement with recent work from *Tiruvadi-Krishnan et al., 2021*, who propose that DNA replication limits division at slow growth but not at fast growth (see also Discussion). However, we note that part of this increase might also be caused by decreasing accuracy of detecting replication initiation during overlapping rounds of replication, which would artificially decrease correlations between replication and division.

*Witz et al., 2019* argued that their single-process model could reconcile adder behavior based on asymmetric cell division (see also their recent comment in *Julou et al., 2020*). For simplicity and analytical tractability, we did not include asymmetric division in the general models shown in *Figure 4C*, but we analyzed its role separately in *Figure 4—figure supplement 2*. We also observed that in the model proposed by *Witz et al., 2019*, asymmetric division drives the inter-division control $\zeta_G$ toward an adder-like process, reaching adder behavior for division asymmetries that are similar to experimentally observed values (*Figure 4—figure supplement 2*). However, this model does not allow $\zeta_{CD}$ to deviate from an adder, thus resulting in a poor agreement upon perturbation of cell width (*Figure 4—figure supplement 5*).

The predictions of *Figure 4C* rely on analytical calculations performed in the limit of small noise. To verify that the levels of cell-to-cell variability would not affect the results, we tested the predictions of our model with simulations at the experimentally observed levels of noise, and as a function of noise levels. *Figure 4—figure supplement 4* shows by direct model simulation that the predictions are robust.

## Discussion

In conclusion, our study suggests that cells control the timing of cell division based on at least two processes in slow-growth conditions: genome replication/segregation and an inter-division process, which relates cell division to size at birth. Accordingly, experimental data obtained in this study and in previous studies are well described by the concurrent-cycles model, while the available single-process models fail to describe our experimental data in unperturbed and perturbed conditions.

Our conclusions are based on the following observations: First, cell size at division and cell size at initiation of DNA replication are correlated in unperturbed cells ($\zeta_{\mathrm{CD}} = 0$, *Figure 3*), as already

observed previously (*Micali et al., 2018b*; *Witz et al., 2019*). Thus, division and replication cannot proceed fully independently of one another, as previously suggested (*Si et al., 2019*). But why can DNA replication alone not account for division control as suggested by *Witz et al., 2019*, in form of an adder between replication initiation and division? When increasing cell width and the average D period with A22, we observed decreasing correlations between DNA replication and division (a decrease of $\zeta_{CD}$ towards -1) (*Figure 3*), which suggests that division becomes decreasingly dependent of replication. At the same time, two other key cell-cycle couplings remained nearly unchanged ($\zeta_G \approx 0$, $\zeta_I \approx 0$). Our data are in line with the idea that a replication-independent process related to size at birth contributes to division control, and that this process is dominant upon width perturbations. Thus, cell division is apparently affected by both cell size at birth and DNA replication.

What is the process that links cell division to size at birth? The concurrent-cycles model suggests that the inter-division process is an adder-like process ($\zeta_H \approx 0$), which shows a mild trend toward sizer with increasing perturbation. The adder-like nature of this process is also supported by experiments with dividing filamentous cells, where DNA replication is likely never limiting cell division (*Wehrens et al., 2018*). Recently, multiple studies suggested that cells divide independently of DNA replication, based on a licensing molecule that accumulates since birth and reaches a critical threshold in copy number at the time of cell septation or division (*Si et al., 2019*; *Zheng et al., 2020*; *Ojkic et al., 2019*; *Harris and Theriot, 2016*; *Panlilio et al., 2021*). The licensing molecules were suggested to be cell-wall precursor molecules (*Harris and Theriot, 2016*), FtsZ or other division-ring components (*Si et al., 2019*; *Ojkic et al., 2019*; *Serbanescu et al., 2020*), or other unknown molecules (*Zheng et al., 2020*). The peptidoglycan accumulation model is based on the assumption that peptidoglycan accumulates in proportion to cell volume, while cell-wall insertion occurs in proportion to cell-surface growth. However, some of us recently demonstrated that cell surface area grows in proportion to biomass (*Oldewurtel et al., 2021*), which makes it more likely that peptidoglycan synthesis and cell-wall insertion happen at equal rates. FtsZ or a different septum component are possible candidates for the inter-division mechanism. Cell size at z-ring formation correlates with total FtsZ abundance (rather than FtsZ concentration) (*Männik et al., 2018*). Furthermore, controlled repression or over-expression of FtsZ delay or accelerate subsequent cell division (*Si et al., 2019*). However, at the same time, the expression of FtsZ is cell-cycle dependent (*Männik et al., 2018*). Whether the accumulation of FtsZ or other divisome components are responsible for an adder-like inter-division process thus requires further investigation.

*Si et al., 2019* recently conducted periodic expression/repression experiments of FtsZ, the mentioned septum component, and DnaA, the major replication-initiation protein, which led them to conclude that replication and division were independent of each other. While their experiments are suggestive of a role of cell size at birth for subsequent cell division, their data do not rule out an additional limiting role of DNA replication for division, which is supported by the adder-like correlations observed between replication initiation and division (*Figure 3*; *Witz et al., 2019*).

How is cell division mechanistically coupled to DNA replication? Z-ring formation and DNA segregation are coupled through the processes of nucleoid occlusion, which inhibits Z-ring formation on top of nucleoids, and *ter* linkage, a process that links the Z-ring to the terminal region of the segregated chromosomes (*Dewachter et al., 2018*). Another link in slow-growth conditions comes from FtsZ expression: FtsZ-protein expression increases in a step-wise manner during the cell cycle (*Männik et al., 2018*), and Z-ring formation happens predominantly after the increase of production (*Männik et al., 2018*). However, which of these or other processes is coupling the timing of replication to division remains to be determined.

Based on the concurrent-cycles model, we predict that inter-division and DNA replication/segregation processes are equally likely limiting cell division ($p_H \approx 0.5$) in two different minimal growth media (*Figure 4—figure supplement 3*), and we previously reported the same balance (*Micali et al., 2018b*) for previous experiments at slow growth (*Adiciptaningrum et al., 2016*; *Wallden et al., 2016*). However, at fast growth, $p_H$ seems to increase, based on fitting our model to data from *Si et al., 2019* (*Figure 4—figure supplement 3*). While it is increasingly challenging to detect the time of initiation accurately in this regime, which could account for part of the increase of $p_H$, support of this trend also comes from a recent study by *Tiruvadi-Krishnan et al., 2021*. They demonstrate that temporal correlations between replication termination and z-ring constriction are high at slow growth, which supports a limiting role of DNA replication for cell division, but

correlations decrease at fast growth, which then requires a different process to control cell division, in qualitative agreement with the concurrent-cycles model (*Figure 4—figure supplement 3*).

The balance between the replication/segregation and inter-division processes at slow growth, over a broad regime of growth rates, is surprising, as it requires that both processes terminate, on average, at the same cell volume $2\langle V_0\rangle$. Under balanced conditions, average cell size after completion of the inter-division and replication/segregation processes are given by $\langle V_0 + \Delta_H\rangle \approx 2\langle\Delta_H\rangle$ and $2\Delta_I 2^{(C+D')/\tau}$ (*Ho and Amir, 2015*), respectively. With $\Delta_I$ constant, $\Delta_H$ must therefore scale in proportion to $2^{[(C+D')/\tau]} \approx 2^{[(C+D)/\tau]}$.

*Zheng et al., 2020* recently re-investigated average cell size and the duration of the C+D period as a function of nutrient-dependent growth rate. While it was previously thought that cell size increases exponentially with growth rate (*Schaechter et al., 1958*), *Zheng et al., 2020* identified a linear relationship. Similarly, they found that the average C+D period shows a Michaelis-Menten-like relationship ($\mathrm{C+D} = \mu/(a\mu + b)$) with average growth rate μ. Based on these experimental findings, they suggested an accumulator model (equivalent to our H-process) that could reconcile the growth-rate dependent increase of average cell size, as long as the threshold molecule was produced at a rate proportional to $1/(\mathrm{C+D})$ on average. Recent theoretical work supports this relationship (*Serbanescu et al., 2020*) based on the assumption of constitutive divisor expression. The same assumption also finds some experimental validation from nutrient-shift data (*Panlilio et al., 2021*). Constitutive divisor-protein expression could provide an explanation for the maintenance of $p_H$ over different unperturbed conditions. However, as soon as only one of the two processes is modulated, for example through width perturbations (*Figure 4*), their balance is broken.

A qualitatively different behavior at slow growth was recently suggested in the already mentioned work by *Tiruvadi-Krishnan et al., 2021*. While they do not put forward a complete cell-cycle model, they suggest that a checkpoint temporally close to DNA replication solely limits the timing of z-ring constriction and therefore cell division at slow growth but not at fast growth. In the future, it will thus be interesting to re-investigate the balance between two different processes by implementing a variant of the concurrent-cycles model that considers an 'and' gate between replication termination and z-ring constriction.

The concurrent-cycles framework assumes that replication initiation is independent of cell division or cell size at birth, based on the robust measurements of adder behavior between subsequent initiations (*Figure 3C*). However, we note that this is not the only possibility, and DNA replication may not be entirely independent of cell division. A complementary hypothesis (*Kleckner et al., 2018*) posits a possible (additional or complementary) connection of initiation to the preceding division event. To test this hypothesis, one could perturb specific division processes by titrating components involved in Z-ring assembly (e.g. titrating FtsZ *Zheng et al., 2016*).

In conclusion, cell-cycle regulation remains to be understood mechanistically. However, from our work it appears that in standard conditions both DNA replication and cell growth since birth play important roles for division timing.

## Materials and methods

### Key resources table

| Reagent type (species) or resource | Designation | Source or reference | Identifiers | Additional information |
|---|---|---|---|---|
| Strain, strain background (*E. coli*) | S233 | This work | NCM3722, *λ::P127-mcherry, dnaN::Ypet-dnaN* | Strain construction |
| Chemical compound, drug | A22 | Cayman Chemicals | 22816-60-0 | |
| Software, algorithm | MATLAB | The MathWorks, Inc. | | |
| Software, algorithm | Oufti | *Paintdakhi et al., 2016* | | |
| Software, algorithm | Schnitzcells | *Young et al., 2012* | | |

### Strain construction

All experiments were carried out with *E. coli* strain S233 (NCM7322, *λ::P-mcherry, dnaN::Ypet-dnaN*). The strain was obtained by a two-step phage transduction into the K-12 strain NCM3722 (wildtype) (*Brown and Jun, 2015*; *Soupene et al., 2003*). First, we introduced mCherry from

MG1655($\lambda$::P127-mcherry,int,kan) (*Vigouroux et al., 2018*) via P1 phage tansduction, then removed integrase and kanamycin-resistance cassette using the pE-FLP system (*St-Pierre et al., 2013*). The resulting strain was transduced with P1 phages lysate of strain S227 (*dnaN::Ypet-dnaN,kan*) (*Reyes-Lamothe et al., 2010*), a kind gift from Rodrigo Reyes-Lamothe. Finally, we removed the kanamycin-resistance cassette using pE-FLP.

## Chemicals

Unless otherwise indicated, all chemicals used in this study were purchased from Sigma-Aldrich. MreB perturbing compound A22 was purchased from Cayman Chemicals and was dissolved in DMSO at a final concentration of 5 mg.mL$^{-1}$. This solution was made every month and stored in small aliquots not defrosted more than two times. An intermediate solution was freshly prepared for each new experiment in the corresponding growth medium.

## Microfluidic chip fabrication

Cell growth was monitored in a microfluidic device for many generations. The device is an adaptation of the mother machine device (*Wang et al., 2010*) with the difference that channels are opened at both ends (*Long et al., 2013*; *Long et al., 2014*). The design of the device was kindly provided by Pietro Cicuta's lab. The chips were replicated from epoxy molds by pouring PDMS (Sylgard 184 with 1:10 w/w ratio of curing agent) and by curing it overnight at 60°C. After cutting the chip and punching inlets (with either a 0.75 mm or 1.5 mm biopsy punch in diameter), the chip was cleaned with scotch tape and bonded to a cleaned glass coverslip (#1.5 24x60 mm). Glass coverslips were cleaned by one hour heated sonication in 2% Helmanex soap, rinsing with water, and then one hour heated sonication in 100% ethanol. The slides were kept in 100% ethanol until used and dried with compressed air just before use. For PDMS bonding to the coverslip, coverslips and PDMS chips were plasma cleaned (Plasma System Cute, Femtoscience), and the assembled chips were baked at 60°C for at least one hour.

Before loading cells, the device's surface was passivated with Pluronic F-127 (P2443, Sigma) at 0.085% final concentration (dissolved in sterile PBS) for 5–30 min at room temperature. The device was then rinsed with growth medium. Loading of the cells was done with no prior centrifugation and with a 5 µm filter attached to the syringe, in order to avoid cells aggregates to clog the channels. All other reagents and media were filtered with a 0.22 µm filter prior to injection in the microfluidic chip. Growth medium flowing in the chip was supplemented with BSA (A9418 Sigma, 10 mg.mL$^{-1}$ final concentration, dissolved in filtered sterile water).

## Growth media

All microscopy experiments were done in M9 minimal medium (*Miller, 1972*) supplemented with 1 mM of MgSO$_4$ (Sigma, M2773) and glycerol (0.2%) as carbon source. If not otherwise indicated we used NH$_4$Cl (19 mM) as nitrogen source. Alternatively, for slower growth, we used Proline (Acros, AC157620250) (10 mM). The composition of M9 minimal medium is: Disodium Hydrogenophosphate (Na$_2$HPO$_4$, S7907, Sigma) (42 mM); Potassium Dihydrogen phosphate (KH$_2$PO$_4$, P0662, Sigma) (22 mM); Sodium Chloride (NaCl 31434, Sigma) (8.6 mM).

## Growth conditions

Bacteria were grown at 37°C. For mother machine experiments, a preculture in the selected M9 growth medium was prepared from a single colony on a LB agar plate after streaking from a glycerol freezer stock. After overnight growth, the culture was back-diluted by a factor 1/50 to 1/100 for growth of 1 to 4 hr at 37°C. The culture was then injected into the mother machine device for population of the channels during one hour without flow. Subsequently, flow with M9 medium (supplemented with A22 if indicated) was started using a syringe pump (Harvard Apparatus). A movie was started at least one hour after starting the flow. We made sure that cells were growing at steady state in terms of growth rate/interdivision time/length/width for at least 6 hr. Any of those quantities were not varying more than 15% during the time course of the experiment (see *Figure 2—figure supplement 3B* for the constancy of growth rate).

For growth rate measurement in liquid culture and snapshots to measure cell dimensions, a preculture was made in the chosen minimal medium from a glycerol stock streak and grown overnight

at 37°C, as above. In the morning, the culture was back-diluted to an OD of 0.005 and treatment with A22 was started. Cells were grown for 1–2 hr at 37°C before growth rate measurements were started. Snapshots were taken after 7 hr of A22 treatment.

## Microscopy

Microscopy was performed on an inverted DeltaVision Elite microscope (GE Healthcare) equipped with a 100X oil immersion phase contrast objective (UPlanSApo 100X NA = 1.4, Olympus). We used a laser-based auto-focusing system to maintain focus on the cells throughout the whole course of the experiment. For fluorescence measurements, we used a Fluorescence light source (Lumencor), a multi-band dichroic beamsplitter (DAPI-FITC-mCherry-Cy5), FITC filter (excitation: 475/28, emission: 525/48) and mCherry filter (excitation: 575/25, emission: 625/45). Parameters for excitation were 10% of light intensity for mCherry, with exposure time of 300 ms and 32% of intensity for YPet, with exposure time of 300 ms. Images were acquired through a sCMOS camera (DV Elite, PCO-Edge 5.5) with an effective pixel size of 65 nm was used, with a frame interval of 6 min for cells grown in M9 ($NH_4Cl$, Glycerol) medium and 8 min for cells grown in M9(Proline, Glycerol) medium. Imaging was done at 37°C in a controlled chamber. Microfluidic flow was controlled with a syringe pump (Harvard Apparatus).

## Image analysis

Image analysis was based on published or custom Matlab scripts. Cells were segmented using the Oufti package (*Paintdakhi et al., 2016*). Dimensions of cells grown in liquid culture and imaged on agarose pads were extracted using Oufti. For cells grown in mother machine channels, we considered all channels that contained cells growing for the whole duration of the experiment. As the cells are trapped in channels and their long axis is aligned with the channel direction, we computed cell length as the distance between the two extreme points of the cell contour (obtained with Oufti) along the channel axis. We subsequently reconstructed cell lineages using the Schnitzcells software (*Young et al., 2012*), and we considered only cells with at least four ancestors for further analysis.

Single-cell growth rate was calculated from an exponential fit to cell length as a function of time. Only cells with positive growth rates and exponential fits with $R^2$ above 0.8 were kept for analysis.

For our statistical analysis of replication-division coupling, we considered triplets of cells (a cell associated with its mother and its grandmother). This allowed us to follow two subsequent replication cycles and the corresponding events of cell division (a C+D period after initiation), even if replication initiation started more than one generation time before division.

To obtain average time points of replication initiation and termination, we generated probability-density maps $p(z/L, t - t_d)$ of finding a DnaN-Ypet spot at a position $z$ along the cell axis (normalized by cell length $L$) at a time $t - t_d$ before cell division (*Figure 2C*). To that end we identified fluorescent spots of Ypet-DnaN: First, a bandpass filter was applied to the YPet fluorescence image (Matlab function *bpass* with 0.8 px and 20 px for the characteristic length scales of noise and objects, respectively). We then considered all local intensity maxima (Matlab function *regionprops*) inside cell contours with peak intensity above a manually defined threshold. We then obtained the average time points of initiation/termination as as inflection points along the x-axis in probability density maps (see *Figure 2C*).

For the detection of DNA-replication initiation and termination in single cells, we did not consider spots but took advantage of the heterogeneous Ypet signal during replication (as illustrated in *Figure 2—figure supplement 1*). After bandpass filtering of the YPet image, we subtracted the median intensity $I_{med}$ for every pixel and took the sum: $I_{tot} = \sum_i (I_i - I_{med})$, where $i$ runs over all pixels inside the cell contour. We divided triplets of cells into two mother-daughter pairs. In each pair, we aimed to identify a complete round of replication that is most recently terminated before before the division of the respective daughter cell. Prior to single-cell analysis, limits for initiation frame and termination frame were obtained from the probability density maps (*Figure 2C*, *Figure 2—figure supplement 1*). Replication/termination was allowed to happen up to 11 time frames (of 6 or 8 min, depending on growth medium) before or after the average time of replication/termination. In each mother-daughter pair, we then identified regions with $I_{tot} > 0$ of a duration of at least 25 min as potential rounds of replication. We then identified the largest region with both initial and final time points within the respective time windows defined above (*Figure 2—figure supplement 1*). We

allowed the D period to be equal to zero if no replication is detected in the two first frames of the two daughter cells. Following this protocol, we identified replication periods in almost all cells (see *Supplementary file 1*).

## Estimation of adder slopes

To measure the added length per ori between subsequent replication initiation events and between replication initiation and subsequent division, respectively, we first calculated an ori-normalized length $L^\star$. To that end, we divided the length of the mother and grandmother cells by two and four respectively. The added length per ori between initiations is then obtained as $\Delta_{\mathrm{I}} = L^\star(t_B^{\mathrm{cell}}) - L^\star(t_B^{\mathrm{mother}})$, irrespectively of whether initiation events happen in cell, mother, or grandmother. Similarly, the added length between replication and subsequent division is obtained as $\Delta_{\mathrm{C+D}} = L^\star(t_d^{\mathrm{cell}}) - L^\star(t_B^{\mathrm{cell}})$. Here, we implicitly assumed symmetric cell division, since division asymmetry is small (5%) in all our experiments (*Figure 4—figure supplement 2*). To test for the influence of division asymmetry on the adder slopes, we corrected the added lengths for the asymmetries of grandmother-mother and mother-cell division events. For example, to correct for the asymmetry in the calculation of the inter-initiation added length, if subsequent initiations happen in mother and daughter cell, we obtain $\Delta_{\mathrm{I}}^{\mathrm{asym}} = \Delta_{\mathrm{I}} + (1 - \alpha)L_d^\star$, where $L_d^\star$ is the ori-normalized length of the mother cell at division, and where $\alpha = \left(L_0^{\mathrm{sibling}} - L_0\right)/\left(L_0^{\mathrm{sibling}} + L_0\right)$ is the division asymmetry between the daughter cell with birth length $L_0$ and its sibling with birth length $L_0^{\mathrm{sibling}}$. Comparing the simple and the more accurate calculation revealed no significant difference for both I and C+D periods, respectively (*Figure 4—figure supplement 2*).

Adder slopes were estimated from a robust fit on the cloud of points using iteratively re-weighted least squares (Matlab, *robustfit* function) to avoid the contribution of occasional outliers. Detailed sample sizes for each experiment are listed in *Supplementary file 1*.

## The use of length fluctuations as a proxy for size fluctuations

For our statistical analysis of cell-cycle progression (*Figure 3*), we used single-cell length fluctuations as a proxy for size fluctuations (rather than fluctuations in volume), for the following reasons:

First, it would be most desirably to measure fluctuations in single-cell mass. Whether fluctuations in surface area or volume are better proxies for mass fluctuations remains to be studied in detail. However, in favor of surface area being a potentially better proxy, we recently showed that the ratio of surface area to mass remains constant during the cell cycle while dry-mass density, the ratio between mass and volume, varies systematically with length (*Oldewurtel et al., 2021*). Cell length, in turn, is directly proportional to surface area $S$ ($S = \pi L W$), independently of polar caps or septum formation, while length and volume show a septum-dependent and non-linear relationship.

Second, in our data, both surface-area and volume calculations are subject to substantial measurement noise in width, so that (within conditions) the best available proxy for mass is actually length. Specifically, relative width variations to be about 10% in the mother machine, but physical cell-to-cell variations are likely about 5% – see our measurements on agarose pads in *Figure 2—figure supplement 2* and (*Oldewurtel et al., 2021*). Hence, while absolute uncertainty in width and length measurements are likely very similar, measurement noise in width leads to much higher uncertainty in volume (by about sixfold in our conditions).

Since cell-to-cell fluctuations in width do not increase with increasing drug concentrations (*Figure 2—figure supplement 2*), we reasoned that the observed decrease of correlations between initiation size and division size (with increasing A22 concentration; *Figure 3H*) is not a consequence of width fluctuations. We also note that our conclusions on size correlations are based on size fluctuations around their respective means, and thus they are not affected by mean-width changes across conditions.

## Mathematical linear-response formalism for adder coupling constants of cell-cycle subperiods

In this section, we present the mathematical framework used in this work to quantify the size control during different cell-cycle subperiods, and to compare experimental results with predictions from different theoretical models. Specifically, this framework provides us with relationships between the

slopes of the different adder plots (*Figure 3*) that must be met by experimental data to support a given model. Thus, the relationships provide a powerful validation/falsification tool for the different models available.

The original formalism presented in *Micali et al., 2018a* is based on the so-called 'size-growth plots' (*Turner et al., 2012*; *Chandler-Brown et al., 2017*; *Grilli et al., 2018*), whose slope ($\lambda$) quantifies the correlation between (logarithmic) size and (logarithmic) multiplicative growth. Here, we adopt an equivalent variant of the formalism based on the slope ($\zeta$) of 'adder plots', which relate the added size over a subperiod to initial size (size at the beginning of the subperiod) (*Jun and Taheri-Araghi, 2015*).

At fast growth, *E. coli* starts DNA replication already in the mother or grandmother, depending on the C+D period and on the generation time ($\langle \tau_{C+D} \rangle > \langle \tau \rangle$). Our framework can take into account such situations for single-process models. However, for the concurrent-cycles model our theory is restricted to non-overlapping rounds of replication/segregation (that is $\langle \tau_{C+D} \rangle > \langle \tau \rangle$). However, we found empirically that the theory also works for overlapping rounds within the range of $\langle \tau_{C+D} \rangle / \langle \tau \rangle$ values observed in our experiments (*Figure 4—figure supplement 4*). For all models, analytical predictions only apply to the limit of small noise and for symmetric division. For comparison with data with overlapping rounds, analysis of the role of noise, and of division asymmetry, we used direct numerical simulations of the models (see the figure supplements to *Figure 4*).

## Standard linear-response formalism based on the slopes of size-growth plots

We recapitulate here the linear-response formalism used in *Micali et al., 2018a*, based on size-growth plots (see also *Amir, 2014*; *Grilli et al., 2018*). This formalism assumes that a genealogy of single cells, whose cell cycles are indexed by $i$, grow exponentially, $V^i(t) = V_0^i e^{\mu^i(t-t_0)}$, where $V_0^i$ and $t_0$ are the cell volume and time at birth, respectively. $V^i(t)$ is the volume of cell cycle $i$ at time $t$, and $\mu^i$ is its growth rate. During a cell cycle, the cell reaches a final size $V_f^i$ in a period of time $\tau^i = t_f - t_0$ (inter-division time), before dividing symmetrically, $V_f^i = 2V_0^{i+1}$.

Since single cells show exponential growth $V_f^i(t) = V_0^i e^{\mu^i \tau^i}$, we decided to expand the logarithmic growth $G_G^i := \mu^i \tau^i$ about its average value ($\langle G_G \rangle \simeq \log 2$) in terms of variations around the logarithmic size at birth $q_0^i := \log V_0^i$. In this way, the size of the newborn cells can be written as

$$2V_0^{i+1} = V_0^i e^{\langle G_G \rangle - \lambda_G \delta q_0^i + \eta_0^i}, \tag{S1}$$

where $\delta q_0^i = \log V_0^i - \langle \log V_0 \rangle \simeq \log V_0^i - \log \langle V_0^i \rangle$ and $\lambda_G$ is the slope of the size-growth plot, which quantifies size homeostasis. Finally, $\eta_0^i$ is assumed to be Gaussian noise with mean zero and standard deviation $\sigma_{q_0}$. This formalism is described in detail in *Amir, 2014*; *Grilli et al., 2018*, and amounts to treating the initial size fluctuations as a linear response problem.

By taking the logarithm of *Equation (S1)*, the variation in logarithmic size of the newborn cell can be expressed as function of the variation of the logarithmic size of the mother cell at birth,

$$\begin{aligned} q_0^{i+1} + \log 2 &= q_0^i + \langle G_G \rangle - \lambda_G \delta q_0 + \eta_0^i \\ q_0^{i+1} + \log 2 - \langle q_0 \rangle &= q_0^i + \langle G_G \rangle - \lambda_G \delta q_0 - \langle q_0 \rangle + \eta_0^i \\ \delta q_0^{i+1} + \log 2 &= \delta q_0^i + \langle G_G \rangle - \lambda_G \delta q_0 + \eta_0^i \\ \delta q_0^{i+1} &= (1 - \lambda_G) \delta q_0^i + \eta_0^i \end{aligned} \tag{S2}$$

Note that $\lambda_G = 1$ corresponds to a sizer since the fluctuation in logarithmic initial size of cell $i+1$ do not depend on the fluctuations in logarithmic size at birth of cell $i$ ($\delta q_0^{i+1} = \eta_0^i$). On the other extreme, $\lambda_G = 0$ corresponds to a timer, in which fluctuation in logarithmic size of cell $i+1$ fully explained by fluctuation in the logarithmic size of the mother cell $i$ ($\delta q_0^{i+1} = \delta q_0^i + \eta_0^i$). $\lambda_G$ can take any intermediate value with $\lambda_G = 0.5$ corresponding to an adder. Multiplying both sides of *Equation (S2)* by the fluctuation in initial logarithmic size $\delta q_0^i$ and taking the average gives us an expression to directly measure the strength of control as a linear-response from data coefficient (*Grilli et al., 2018*),

$$(1 - \lambda_G) = \frac{\langle \delta q_0^{i+1} \delta q_0^i \rangle}{\sigma_{q_0}^2} \, . \tag{S3}$$

The same formalism can be used to estimate the strength of size control over subperiods (notably, the C+D period) and between consecutive initiation events (I period) (*Micali et al., 2018a*). Hereafter, the quantities $q_X^i$ refer to the logarithmic volume at cell cycle progression stage $X$ of the cycle $i$. We consider for instance the size-growth coupling during the $C + D$ period in the simple case in which initiation and termination both happen in the cell $i$, and we write the following expressions to relate size fluctuations before and after this subperiod

$$\begin{aligned}
q_0^{i+1} + \log 2 &= q_B^i + \langle G_{C+D} \rangle - \lambda_{C+D} \delta q_B + \eta_B^i \\
q_0^{i+1} + \log 2 - \langle q_0 \rangle + \langle q_0 \rangle &= q_B^i + \langle G_{C+D} \rangle - \lambda_{C+D} \delta q_B - \langle q_B \rangle + \langle q_B \rangle + \eta_B^i \\
\delta q_0^{i+1} &= \delta q_B^i - \lambda_G \delta q_0 + \langle G_{C+D} \rangle - \log 2 - \langle q_0 \rangle + \langle q_B \rangle + \eta_B^i \\
\delta q_0^{i+1} &= (1 - \lambda_{C+D}) \delta q_B^i + \eta_B^i,
\end{aligned} \tag{S4}$$

where the log-size fluctuation at initiation is $\delta q_B^i := q_B^i - \langle q_B \rangle \approx \log(V_B^i/\langle V_B \rangle)$, with $V_B$ size at initiation, and $\eta_B^i$ Gaussian noise with mean zero and standard deviation $\sigma_{q_B}$. In the case in which DNA replication starts in the mother (cycle $i$) and terminates in a subsequent cell cycle (in daughters: $n = 2$, in granddaughters: $n = 3$), *Equation (S4)* becomes $\delta q_0^{i+n} = (1 - \lambda_{C+D}) \delta q_B^i + \eta_B^i$. In the same way, one can represent the control strength for the $I$ and $B$ period (*Micali et al., 2018a*) by the following expressions linking logarithmic cell size fluctuations before and after the subperiods,

$$\delta q_B^{i+1} = (1 - \lambda_I) \delta q_B^i + \eta_B^i. \tag{S5}$$

$$\delta q_B^i = (1 - \lambda_B) \delta q_0^i + \eta_0^i. \tag{S6}$$

## From size-growth plots to adder plots

As for $\lambda_G$, the control parameters $\lambda_X$ calculated from logarithmic volumes quantify size homeostasis. For small size fluctuations, they are in 1:1 relation with the slopes of the corresponding adder plots *Grilli et al., 2017*. Here, we translate the $\lambda$-formalism to the slopes of adder plots $\zeta_X$ (*Jun and Taheri-Araghi, 2015*). *Equation (S1)* can be rewritten as

$$2V_0^{i+1} = Q_G (V_0^i)^{1-\lambda_G} \langle V_0 \rangle^{\lambda_G} + \nu_0^i \, , \tag{S7}$$

where $Q_G = e^{\langle G_G \rangle} = \exp\langle \log V_f/V_0 \rangle$, and $\nu_0^i$ is the Gaussian noise with mean zero and standard deviation $\sigma_{V_0}$. *Equation (S7)* was first introduced in *Amir, 2014*. Following this study, expanding around the average size, for small fluctuations (*Amir, 2014*; *Grilli et al., 2017*) we obtain a mapping between added size and slope of the size-growth plot,

$$\begin{aligned}
2V_0^{i+1} &= Q_G \langle V_0 \rangle + (1 - \lambda_G) Q_G \delta V_0^i + \nu_0^i \\
2V_0^{i+1} - V_0^i - \langle V_0 \rangle &= Q_G \langle V_0 \rangle + [(1 - \lambda_G) Q_G - 1] \delta V_0^i - 2 \langle V_0 \rangle + \nu_0^i \\
\delta \Delta_G^i &= +[(1 - \lambda_G) Q_G - 1] \delta V_0^i + \nu_0^i.
\end{aligned} \tag{S8}$$

Here $\Delta_G^i = V_f^i - V_0^i$ is the added size during a cell cycle, and $\delta \Delta_G^i = \Delta_G^i - \langle \Delta_G^i \rangle$ is its fluctuation. Hence, by definition, the term in square brackets must be the slope of the adder plot

$$\zeta_G := (1 - \lambda_G) Q_G - 1. \tag{S9}$$

Solving the equation for $\lambda_G$, we get

$$(1 - \lambda_G) = \frac{(\zeta_G + 1)}{Q_G}, \tag{S10}$$

which can be used (assuming small fluctuations *Grilli et al., 2017*) to convert the slope $\zeta_G$ of the adder plot into the slope of the size-growth plot $\lambda_G$, and vice versa.

It is straightforward to extend the relationship to cell-cycle subperiods and to the inter-initiation period, leading to the following relationships

$$\zeta_{C+D} := (1 - \lambda_{C+D})Q_{C+D} - 1 \tag{S11}$$

$$\zeta_B := (1 - \lambda_B)Q_B - 1 \tag{S12}$$

$$\zeta_I := (1 - \lambda_I)Q_I - 1 \,, \tag{S13}$$

where $\quad Q_{C+D} = \exp\langle\log 2^n V_0/V_B\rangle, \qquad Q_B = \exp\langle\log V_B/(n V_0)\rangle, \qquad Q_I = \exp\langle\log n V_B^{i+1}/V_B^i\rangle \quad$ and $n = \lfloor \tau_{C+D}/\tau \rfloor + 1$.

It is important to notice that for inter-division and inter-initiation events in symmetrically dividing cells $Q_{G,I} \simeq 2$. For these subperiods, adder behavior is equivalent to $\zeta_{G,I} = 0$. However, the same equivalence does not hold for other subperiods, and in particular of the $B$ and $C + D$ period, of interest here, since $Q_{B,C+D} \neq 2$.

## Adder coupling constants for single-process ICD models

We call here 'ICD' models all single-process models that assume a cell-size-independent mechanism in control of the inter-initiation process (I period) and a mechanism that couples cell division to the size of DNA replication initiation (C+D period). We already generalized the approach of *Ho and Amir, 2015*; *Witz et al., 2019* to arbitrary coupling constants for the $C + D$ period (*Micali et al., 2018a*). In this class of models, DNA replication is the limiting process setting subsequent division and initiation events. This section presents the generalized relationships for ICD models in the formalism of adder coupling constants, for non-overlapping and overlapping replication rounds, used in *Figure 4* of the main text and its supplements.

From *Equation (S8)* and the equivalent equations for $C + D$, $B$ and $I$, we can derive the following relationships

$$\delta V_0^{i+1} = \frac{(1-\lambda_G)Q_G}{2}\delta V_0^i + \nu_0^i = \frac{(\zeta_G + 1)}{2}\delta V_0^i + \nu_0^i \tag{S14}$$

$$\delta V_B^{i+1} = \frac{(1-\lambda_I)Q_I}{2}\delta V_B^i + \nu_B^i = \frac{(\zeta_I + 1)}{2}\delta V_B^i + \nu_B^i \tag{S15}$$

$$\delta V_B^i = (1-\lambda_B)Q_B\delta V_0^i + \nu_0^i = (\zeta_B + 1)\delta V_0^i + \nu_0^i \tag{S16}$$

$$\delta V_0^{i+n} = \frac{(1-\lambda_{C+D})Q_{C+D}}{2n}\delta V_B^i + \nu_B^i = \frac{(\zeta_{C+D} + 1)}{2n}\delta V_B^i + \nu_B^i \,, \tag{S17}$$

where $i + n$ generalizes to the case in which the size at birth of cell $i + n$ by replication initiation in cell $i$.

In ICD models, the coupling constants $\zeta_I$ and $\zeta_{C+D}$ are treated as input control parameters, while $\zeta_G$ and $\zeta_B$ are outcomes of the model, measured as observable correlations. The predicted correlations for ICD models are (see *Micali et al., 2018a*),

$$\begin{cases} (\zeta_G + 1) = \frac{1}{(2n)^2}(\zeta_{C+D} + 1)^2(\zeta_I + 1)\frac{\sigma_{V_B}^2}{\sigma_{V_0}^2} \\ (\zeta_B + 1) = \frac{1}{2^{(n+1)n}}(\zeta_{C+D} + 1)(\zeta_I + 1)^n\frac{\sigma_{V_B}^2}{\sigma_{V_0}^2} \end{cases} \tag{S18}$$

The model by Witz and coworkers presented in *Witz et al., 2019* falls in this broad category, with the assumption that $\zeta_{I,C+D} = 0$, i.e. the coupling constants impose perfect adders both between initiation events and during the C+D period. The predicted correlation patterns for this model are

$$\begin{cases} (\zeta_G + 1) = \frac{1}{(2n)^2}\frac{\sigma_{V_B}^2}{\sigma_{V_0}^2} \\ (\zeta_B + 1) = \frac{1}{2^{(n+1)n}}\frac{\sigma_{V_B}^2}{\sigma_{V_0}^2}. \end{cases} \tag{S19}$$

Note that although the model presented in *Witz et al., 2019* falls in the broad category of ICD models, the authors of this study extend the model with an additional parameter, accounting for asymmetric division. This additional ingredient allows their theory to deviate from the predictions of *Equation (S19)*. *Figure 4—figure supplement 2* illustrates this point. As discussed in the main text, asymmetric division can drive $\zeta_G$ toward adder behavior. However, in our hands this requires unrealistically high values of asymmetry. Furthermore, this model fails to reproduce the results of the A22 perturbation presented in this work, since the specific $C + D$ control pattern is postulated in the model, while it changes with the perturbation in the experiments (*Figure 4C* in the main text).

## Concurrent cycles

This section presents the predicted correlation patterns for the concurrent cycles framework in terms of adder coupling constants. In this model (*Micali et al., 2018a*), two cycles are in competition for setting cell division. According to the size-growth framework, a cycle '$H$' starts from cell division, and has control strength $\lambda_H$ over the next division event. In addition, a cycle '$C + D'$' starts from initiation of DNA replication and has control strength $\lambda_{C+D'}$ over the the division event following termination of DNA replication and segregation. At the single-cell level, the slowest process set division, and the parameter $p_H$ encodes the average probability of the cycle $H$ to set division.

In the concurrent cycles model, the control strength of the inter-division process ($H$), of the inter-initiation process ($I$), and of the replication-segregation processes set by initiation ($C + D'$) are inputs of the model. Following *Micali et al., 2018a*, the latter is assumed to be a pure timer, i.e. $\lambda_{C+D'} = 0$. In contrast, the slopes resulting from the competition of the two concurrent cycles, that is the inter-division ($G$) slope and the slopes over the $C + D$ period are outcomes of the model, that is, predictions that can be validated using experimental data.

Following a similar approach to *Micali et al., 2018a* and using *Equations S14-S17*, we obtain

$$\langle \delta V_0^{i+1} \delta V_0^i \rangle = \frac{(\zeta_G + 1)}{2}\sigma_{V_0}^2 = p_H \frac{(\zeta_H + 1)}{2}\sigma_{V_0}^2 + (1 - p_H)\frac{Q_{C+D'}}{2}(\zeta_B + 1)\sigma_{V_0}^2,$$
$$\langle \delta V_B^i \delta V_0^i \rangle = (\zeta_B + 1)\sigma_{V_0}^2 = \frac{(\zeta_{C+D} + 1)}{2}\frac{(\zeta_I + 1)}{2}\sigma_{V_B}^2,$$
$$\langle \delta V_0^{i+1} \delta V_B^i \rangle = \frac{(\zeta_{C+D} + 1)}{2}\sigma_{V_B}^2 = p_H\frac{(\zeta_H + 1)}{2}(\zeta_B + 1)\sigma_{V_0}^2 + (1 - p_H)\frac{Q_{C+D'}}{2}\sigma_{V_B}^2,$$

where the effective parameter $p_H$ quantifies the probability that the inter-division process is limiting, and is a function of basic parameters that are fixed in a given condition, such as mean size at initiation and noises (see *Micali et al., 2018a*).

The above equations can be recast into the following relationships involving adder coupling constants:

$$\begin{cases} (\zeta_G + 1) = p_H(\zeta_H + 1) + (1 - p_H)Q_{C+D'}(\zeta_B + 1) \\ (\zeta_B + 1) = (\zeta_{C+D} + 1)(\zeta_I + 1)\frac{\sigma_{V_B}^2}{4\sigma_{V_0}^2} \\ (\zeta_{C+D} + 1) = \frac{(1 - p_H)Q_{C+D'}}{\left(1 - p_H\frac{(\zeta_H + 1)(\zeta_I + 1)}{4}\right)} \ . \end{cases} \tag{S20}$$

Finally, for the specific case of the adder-adder model in which both the inter-initiation and the $H$ processes are adders ($\zeta_I = 0$ and $\zeta_H = 0$), the same relationships simplify into the following scheme,

$$\begin{cases} (\zeta_G + 1) = p_H + (1 - p_H)Q_{C+D'}(\zeta_B + 1) \\ (\zeta_B + 1) = \frac{(1 - p_H)}{\left(1 - \frac{p_H}{4}\right)}\frac{Q_{C+D}\sigma_{V_B}^2}{4\sigma_{V_0}^2} \\ (\zeta_{C+D} + 1) = \frac{(1 - p_H)Q_{C+D'}}{\left(1 - \frac{p_H}{4}\right)} \ . \end{cases} \tag{S21}$$

Note that *Equations (S20)-(S21)* are valid for $n = 1$, that is for initiation and termination that happen in the same cell cycle. As discussed in *Micali et al., 2018a* simulations are used to extend the results to $n > 1$.

The latter model involving adders over $I$ and $H$ is used for the comparison in *Figure 4* of the main text, while a more general model fixing $\zeta_I = 0$ but allowing $\zeta_H$ to vary is used for the fit in *Figure 4—figure supplement 1*. Note that in the above expressions $Q_{C+D'}$ is the growth during the $C + D'$ period and is not measurable directly. To bypass this problem, we approximate it by

$Q_{C+D'} = 1.8$, which is the average measured $Q_{C+D}$ in unperturbed conditions. $Q_{C+D'}$ is equal to $Q_{C+D}$ for $p_H = 0$. In unperturbed conditions, where $p_H \simeq 0.5$, $Q_{C+D'} \leq Q_{C+D}$, and the two values are similar, since they differ only by the low-CV noise of the inter-division process. For the A22 perturbations, we assumed that the value of $Q_{C+D'}$ remains constant, as the $C + D'$ period should be unperturbed by A22 increasing concentrations (as supported by *Figure 2C*, since the measurable $C$ period is on average constant). We also note that this approximation is equivalent to the reasonable assumption that $Q_H \simeq 2$ used in *Micali et al., 2018a*.

## Brief description of simulations

In this manuscript, we used stochastic simulations for two reasons: (i) to explore the role of asymmetric division in ICD models (*Figure 4—figure supplement 2*), as suggested by *Witz et al., 2019*, (ii) to validate the analytical predictions for $\zeta_G$ and $\zeta_{CD}$ for the concurrent cycles model and in particular the robustness of the small noise approximation and to quantitative extend concurrent cycle predictions for $\langle \tau_{C+D} \rangle / \langle \tau \rangle > 1$ (*Figure 4—figure supplement 4*).

For simulations in *Figure 4—figure supplement 2* that account for asymmetric division, we were inspired by the model in *Witz et al., 2019*. Briefly, for each initiation event $V_B^i$, the number of origins $n_{oris}$ is duplicated and two random added lengths are chosen from log-normal distributions for the I-period ($\Delta_I^i$) and the C+D-period ($\Delta_{CD}^i$), respectively. Note that both means ($\langle \Delta_I \rangle$ and $\langle \Delta_{CD} \rangle$) and standard deviation ($\sigma_I$ and $\sigma_{CD}$) of the distributions are parameters inferred from data. $\Delta_{CD}^i$ sets the division event: $V_d^i = V_B^i + \Delta_{CD}^i$, if $n_{oris} = 1$, if $n_{oris} = 2$. Events with $n_{oris} > 2$ are rare in the conditions used in *Figure 4—figure supplement 2*. However, the simulations can account for those events correcting for asymmetries in the multiple divisions and ensuring an added size between $V_B^i / 2^{n_{oris}-1}$ and the triggered division event equal to $\Delta_{CD}^i$. The number or origins is divided by two at each division event. To account of asymmetric division, the newborn cell has volume $V_0^{i+1}$ set by a Gaussian random variable with mean $V_d^i/2$ and standard deviation $\alpha V_d^i/2$. Typical values of $\alpha$ from our experimental data are 0.05 (see *Figure 4—figure supplement 2*). The next initiation event is set by $\Delta_I^i$ if the next initiation event is in the same cell cycles, $V_B^{i+1} = \frac{V_B^i}{2} + n_{oris}\Delta_I^i - \left(\frac{V_d^i}{2} - V_0^{i+1}\right)$ if the next initiation event is in the following cell cycle. More complicated scenarios in which the next initiation event is in further cell cycles accounts for the multiple asymmetric division events and calculate the actual added size. In the conditions used in *Figure 4—figure supplement 2* these events are rare.

For simulations in *Figure 4—figure supplement 4* of the concurrent-cycles model with perfectly symmetric division, we refer the reader to *Micali et al., 2018b*.

## Description of the analysis of data from the literature

To compare our findings with data available in the literature, we downloaded data of untreated conditions from *Si et al., 2019* (downloaded at https://www.sciencedirect.com/science/article/pii/S0960982219304919) and *Witz et al., 2019* (downloaded at https://zenodo.org/record/3149097#.X7PKA9NKhBx). Excel files are imported in MATLAB using the function *readtable* and all the subsequent analysis has been performed with MATLAB. Since this manuscript is focused on slow-growth conditions, we restrict our comparision with (*Si et al., 2019*) to their slow-growth conditions (*MG1655 acetate* and *NCM3722 MOPS arginine*) (see *Figure 4—figure supplement 3*).

Note that *Si et al., 2019* report the initiation size per origin without specifying the number of origins and without providing the added size during *C+D*. For this reason, we assume the number of origins by plotting the size at initiation vs the size of newborn cells. Cells in which the initiation per origins is smaller than the size at birth are considered to terminate DNA replication in the daughter cell. In this case, the added size during *C+D* is estimated from ( *division size (micron) - initiation size per ori (micron)/ 2 + division size (micron) daughter - newborn size (micron) daughter*. Cells in which the initiation per origins is larger than the size at birth are considered to terminate DNA replication in the same cycle as initiation started. Hence, the added size during *C+D* is *division size (micron) - initiation size per ori (micron)*. The added size between division events is estimated from *division size (micron) - newborn size (micron)*. The added size during two consecutive initiation events is estimated from ( *division size (micron)-initiation size per ori (micron) ) / 2 +initiation size per ori (micron) daughter - newborn size (micron) daughter*). The slopes $\zeta_G$, $\zeta_I$ and $\zeta_{CD}$ were calculated by applying

the *robustfit* function in MATLAB to the clouds of points of the inter division, inter initiation and C +D adder plots, respectively.

The data from *Witz et al., 2019* are in a different format which provides the inter-division, inter-initiation and $C + D$ added size as well as size at birth and size at initiation. For this reason, we were able to calculate the $\zeta_G$, $\zeta_I$, and $\zeta_{CD}$ directly using the added quantities and the *robustfit* function in MATLAB.

## Acknowledgements

We thank the Biomaterials and Microfluidics core facility of Institut Pasteur, and particularly Samy Gobaa, for the help in the fabrication of microfluidic chips, former student Ewen Corre for first investigations of DnaN-Ypet behavior, members of the Microbial Morphogenesis and Growth lab for discussion and technical assistance, Rodrigo Reyes-Lamothe for advice and for sharing the DnaN-Ypet-containing strain. We also thank Pietro Cicuta and Mia Panlilio for sharing microfluidic devices and advice about microfluidic protocols. This work was supported by grants to SVT from the European Research Council (ERC) under the Europe Union's Horizon 2020 research and innovation program [Grant Agreement No. (679980)], the French Government's Investissement d'Avenir program Laboratoire d'Excellence 'Integrative Biology of Emerging Infectious Diseases' (ANR-10-LABX-62-IBEID), the Mairie de Paris 'Emergence(s)' program, and the Volkswagen Foundation. MCL was funded by the Italian Association for Cancer Research AIRC-IG (REF: 23258). GM was supported by grant Nr. 310030-188642 from the Swiss National Science Foundation to Martin Ackermann.

## Additional information

### Funding

| Funder | Grant reference number | Author |
|---|---|---|
| H2020 European Research Council | 679980 | Sven van Teeffelen |
| Agence Nationale de la Recherche | ANR-10-LABX-62-IBEID | Sven van Teeffelen |
| Université Paris Descartes | | Sven van Teeffelen |
| Volkswagen Foundation | | Sven van Teeffelen |
| Italian Association for Cancer Research | 23258 | Marco Cosentino Lagomarsino |
| National Science Foundation | 310030_188642 | Gabriele Micali |

The funders had no role in study design, data collection and interpretation, or the decision to submit the work for publication.

### Author contributions

Alexandra Colin, Conceptualization, Data curation, Software, Validation, Investigation, Methodology, Writing - original draft, Writing - review and editing; Gabriele Micali, Conceptualization, Data curation, Software, Formal analysis, Validation, Investigation, Methodology, Writing - original draft, Writing - review and editing; Louis Faure, Software, Methodology; Marco Cosentino Lagomarsino, Conceptualization, Methodology, Writing - original draft, Project administration, Writing - review and editing; Sven van Teeffelen, Conceptualization, Software, Funding acquisition, Validation, Investigation, Methodology, Writing - original draft, Project administration, Writing - review and editing

### Author ORCIDs

Alexandra Colin https://orcid.org/0000-0002-9144-3282
Louis Faure https://orcid.org/0000-0003-4621-586X
Marco Cosentino Lagomarsino https://orcid.org/0000-0003-0235-0445
Sven van Teeffelen https://orcid.org/0000-0002-0877-1294

Decision letter and Author response
Decision letter https://doi.org/10.7554/eLife.67495.sa1
Author response https://doi.org/10.7554/eLife.67495.sa2

# Additional files

## Supplementary files

• Supplementary file 1. Characterization of different single-cell datasets. Including **Experiment Name, strain, growth medium, perturbation, Numberofcells:** number of cells taken into account, **Percentagekept:** Percentage of cells kept after the DNA replication scoring, that is, cells in which we were able to detect initiation and termination, **Growthrate, GrowthrateCV:** exponential growth rate of the cell (obtained from an exponential fits on single-cell length; units: min$^{-1}$) and CV (coefficient of variation), **Width:** Mean cell width (units: μm), **BirthLength, BirthLengthCV:** Mean length at birth (units: μm) and CV, **InitiationLength, InitiationLengthCV:** Mean initiation length per ori (units: μm) and CV, **DivisionLength, DivisionLengthCV:** Mean length at division (units: μm) and CV, **taucycle, taucycleCV:** Mean duration of the cell cycle (units: min) CV, **tauI, tauI_CV:** Mean duration of the inter-initiation time (units: min) and CV, **tauC, tauC_CV:** Mean C period (units: min) and CV, **tauD, tauD_CV:** Mean D period (units: min) and CV, **slope_division_adder, slope_initiation_adder, slope_CD_adder:** Slopes of the division-adder plot, initiation-adder plot, and C+D-adder plots, respectively.

• Supplementary file 2. Single-cell data used in this study. **ExperimentName:** Label of the experiment (same as in the *Supplementary file 1*), **grandmother:** Cell ID of the grandmother cell, **mother:** Cell ID of the mother cell, **cell:** Cell ID of the cell, **Linit_cell:** Initiation length per ori of the cell (μm) (note that this initiation can happen in the mother cell), **Linit_mother:** Initiation length per ori of the mother (μm) (note that this initiation can happen in the grandmother cell), **Lterm_cell:** Termination length per ori of the cell (μm), **Lterm_mother:** Termination length per ori of the mother (μm), **Tinit_cell:** Initiation time of the cell (min), **Tinit_mother:** Initiation time of the mother (min), **Tterm_cell:** Termination time of the cell (min), **Tterm_mother:** Termination time of the mother (min), **Tbirth_grandmother:** Birth time of the grandmother (min), **Tbirth_cell:** Birth time of the cell (min), **Tbirth_mother:** Birth time of the mother (min), **Tdivision_mother:** Division time of the mother (min), **growthrate_mother:** Growth rate of the mother cell (min$^{-1}$), **growthrate_grandmother:** Growth rate of the grandmother cell (min$^{-1}$), **Tdivision_cell:** Division time of the cell (min), **growthrate_cell:** Growth rate of the cell (min$^{-1}$), **Lbirth_cell:** Birth length of the cell (μm), **Ldivision_cell:** Division length of the cell (μm), **Width_cell:** Width of the cell (μm).

• Transparent reporting form

## Data availability

All data generated or analysed during this study are included in supplemental datasets provided for each figure. Average quantities and sample sizes for each biological replicate can be found in Supplementary file 1. Supplementary file 2 contains all single-cell data used in this study.

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
