## [Decision Letter]

**Acceptance summary:**

How the bacterium *E. coli* decides when to divide is an interesting, important, unsolved and highly controversial topic of interest to readers across disciplines, including microbiology, cell biology and statistical physics. Popular "single process" models invoke regulation at the step of replication initiation or at the step of cell division per se, whereas these authors have previously proposed a "concurrent cycles" model in which both processes are relevant, with different prominences in different situations. Consistent with the authors' motivating hypothesis, in the particular perturbed condition investigated in this work, a process different from DNA replication becomes increasingly important for division control as the degree of perturbation increases, which provides a new challenge to models for cell division control.

**Decision letter after peer review:**

Thank you for submitting your article "Two different cell-cycle processes determine the timing of cell division in *Escherichia coli*" for consideration by *eLife*. Your article has been reviewed by 3 peer reviewers, and the evaluation has been overseen by a Reviewing Editor and Naama Barkai as the Senior Editor. The following individuals involved in review of your submission have agreed to reveal their identity: Nancy Kleckner (Reviewer #2); Chenli Liu (Reviewer #3).

Essential revisions:

(1) Please include a discussion to address the possibility that DNA replication and cell division may not be entirely independent.

You could additionally perform experiments to strengthen your conclusions, now or at a later time – but this is not a requirement, it is entirely up to you. Suggestions: perturb specific cell division processes by titrating components involved in Z-ring assembly (e.g., titrating FtsZ as in Zheng et al. PNAS 2016). Alternatively, following the authors' reasoning, perturbation to DNA replication should exaggerate the effect of the replication-dependent processes on division timing. One possible approach is to slow down the speed of replication fork as done in Zhu et al. mBio 2017 or Si et al. Curr. Biol. 2017. It is important to see if the single-cell correlations can be restored when replication is perturbed in A22 treated cells.

2) Please edit the manuscript to address all points raised in the "Recommendation for authors" sections below.

*Reviewer #2 (Recommendations for the authors):*

Overall, this paper is well and clearly written.

Figure 2.

A. Are the two cells outlined below the same as the ones in the image series above? Does not seem so. There are too many "time points" in the below cartoon vs the above images and the positions of the spots don't correspond. This is hard to understand.

B. It seems that cell lengths are greater in liquid culture than in mother machine at all A22 concentrations; the same might or might not be true of width. Why? Does this matter? Is it completely clear that the cells are in steady state after 6 hours (see discussion in Zheng et al., 2020)?

C. (i) Can the authors explain why, in the unperturbed case, the DnaN signal seems to "split" into two parts and then become single again? This does not happen in the A22 samples. Does this matter?

(ii) Also, it is noted that the times defined by these images do not "strictly agree" with average replication/termination times…Please explain? Are we supposed to see these as illustrating the general pattern of the change in replication initiation time, vs division, and the fact that C period is the same in all conditions, without worrying about the exact length of the C-period? Or…how are the average replication/termination times actually determined (vs these images)?

(iii) Also – it might be useful to indicate the "C" and "D" periods explicitly, eg in the unperturbed case? C period is time between red and yellow dotted lines and D period is time between yellow dotted line and next cell division, right? But in the A22 case, C period spans a division and D-period is again between yellow line and next division… This is obvious to people used to C+D but is something to keep in mind in terms of the biology.

Figure 3.

– Legend has typo: B,D,F: Slopes of the added sizes corresponding to A, C, D should be "E", respectively

– In Panel E

(i) There is an assumption involved in plotting the "average length added PER ORIGIN in C+D" versus the "initiation length PER ORIGIN". (i.e. in normalizing "per origin").

The rationale for this approach should be made explicit. It is presumably derived from the Witz-type analysis in (C) which considers that everything scales with the number of origins. Perhaps this is in the SI/Methods, but it needs to be explained in the text.

(ii) Does it matter that in A22-treated cells, the C-period is spread over two division cycles? Probably not, since the C period is constant in all conditions, but this might be mentioned?

Figure 4.

Panel AB. This is confusing (at least to this reviewer). In Panel A, the cartoon and legend say: "Two independent inter-division and timer-like replication/segregation processes with control parameters ζH and ζCD¨ = 1, respectively, must be completed before division occurs. The adder-like inter-initiation processes with control parameter ζI = 0 determines size at initiation." But in Panel B: "Model-fitting to experimental data reveals the probability pH of the inter-division process to control cell division as a function of increasing D period (with increasing A22 concentration), assuming constant control parameters ζH = 0, ζC+D¨ = 0, ζI = 0."

Why does the model in (A) use different values for ζH and ζCD¨ than those used for the model fitting?

*Reviewer #3 (Recommendations for the authors):*

1. It has been reported that MreB affected chromosome segregation (see Ram Madabhushi and Kenneth J. Marians, Molecular Cell, 2009, and Thomas Kruse et al., EMBO J, 2003 for details). The authors need to put this issue into discussion and/or analysis.

2. In the model, if QC+D' keeps constant, it is natural to conclude that 'inter-division process' would become more limiting (i.e., pH would increase), in the case of increased C+D periods. Since C+D' is unmeasurable, more discussion are need to explain the contribution of the experimental data to the conclusion of the model.

3. In Figure 3F, it would be helpful to show a line indicating 'always depends on DNA replication'.

4. Line 33, should be Figure 1BC.

5. Line 109-111, YPet-DnaN is used to measure periods of DNA replication. Since DnaN is reported to be associated with DNA for some time after replication is done (M. Charl Moolman et al. Nature Communications, 2014), one may overestimate the C-period in this manner.

6. There are three control parameters (𝜁H, 𝜁I and 𝜁CD′) in the model. The first two were assigned to 0 with explanation in Line 200-210. It needs more explanation of 𝜁CD′ in the main text.

7. In Figure 2B and 2C, standard deviations for each gray rectangle are missing.

[Editors' note: further revisions were suggested prior to acceptance, as described below.]

Thank you for resubmitting your work entitled "Two different cell-cycle processes determine the timing of cell division in *Escherichia coli*" for further consideration by *eLife*. Your revised article has been reviewed by 3 peer reviewers and the evaluation has been overseen by Naama Barkai as the Senior Editor, and a Reviewing Editor.

The manuscript is essentially ready for publication. Please address the two remaining comments from referee 3 as you see fit, for example by editing the two figures as suggested by the referee.

*Reviewer #1 (Recommendations for the authors):*

Revisions are satisfactory. I have no further comment.

*Reviewer #2 (Recommendations for the authors):*

All concerns of this reviewer have been fully addressed in the revised manuscript.

*Reviewer #3 (Recommendations for the authors):*

The revision makes the manuscript much more comprehensible. I have two remaining comments:

1. Since most cell size-related models emphasize steady-state, providing further evidence that the single-cell data here were acquired during steady-state is important, as also commented by Reviewer 1. Similar to Figure 2-Supplement 2, I would recommend plotting the 'inter-division time' vs. 'Time of birth' as well, because growth rate is likely to be constant with continuous medium supply but 'inter-division time' may not, whereas steady-state requires a similar rate for both cell growth and division. Furthermore, showing these two plots for the highest A22 dosage (1 ug/mL) data or even all dosages used in the work would be a very strong evidence that the cells were in steady-state.

2. 'Initiation mass' (or initiation volume) is commonly used to refer cell size upon initiation. With altered cell width in this work, cell length is no longer a good proxy for cell mass, especially in important figures like Figure 3E. therefore, I strongly recommend to use cell volume in both axis in Figure 3E (or entire Figure 3) to precisely convey the message that the added mass becomes negatively correlated with initiation mass. The dots in Figure 3E (right panel) may also be less clustered and the trend line could be more significant, if volume is used there.

---

## [Author Response]

Essential revisions:1) Please include a discussion to address the possibility that DNA replication and cell division may not be entirely independent.You could additionally perform experiments to strengthen your conclusions, now or at a later time – but this is not a requirement, it is entirely up to you. Suggestions: perturb specific cell division processes by titrating components involved in Z-ring assembly (e.g., titrating FtsZ as in Zheng et al. PNAS 2016). Alternatively, following the authors' reasoning, perturbation to DNA replication should exaggerate the effect of the replication-dependent processes on division timing. One possible approach is to slow down the speed of replication fork as done in Zhu et al. mBio 2017 or Si et al. Curr. Biol. 2017. It is important to see if the single-cell correlations can be restored when replication is perturbed in A22 treated cells.

We have added to the discussion the possibility that DNA replication could depend on cell division, and that further testing with different perturbations is needed to clarify this point. Specifically, we now write:

‘The concurrent-cycles framework assumes that replication initiation is independent of cell division or cell size at birth, based on the robust meqasurements of adder behaviour between subsequent initiations (Figure 3C). […].To test this hypothesis one could perturb specific division processes by titrating components involved in Z-ring assembly (e.g., titrating FtsZ \Zheng et al., 2016).’

While we acknowledge that the experiments proposed by the reviewers would likely be very informative for an even better understanding of cell-cycle regulation, we decided not to conduct those at this point.

2) Please edit the manuscript to address all points raised in the "Recommendation for authors" sections below.

Please see our point-by-point response below. Furthermore, we have made a few changes to the manuscript in response to a recent preprint by the labs of Jaan Mannik and Ariel Amir (https://www.biorxiv.org/content/10.1101/2021.02.18.431686v1). This article also addresses the question of how DNA replication affects cell division in *E. coli* during slow growth. Their manuscript supports our general finding that cell division is influenced by both DNA replication and by a replication-independent process, depending on growth conditions. Complementary to our work, they find that the influence of DNA replication depends on average growth rate: It is strongest at slow growth and reduced at faster growth, in qualitative agreement with predictions from our work, which, in turn, is based on our own data and on the previously published data by Witz et al. 2019 and Si et al. 2019. In response to their manuscript, we now (a) added more information on the limiting nature of DNA replication for different growth rates according to our data and model, and according to previously published data by Witz et al. 2019 and by Si et al. 2019, and (b) added a short discussion of the paper to the Discussion.

a. We included a comparison of p_H_ across different average doubling times, based on previously published data and our own datasets in two different growth media. We now write:

‘We also verified that the concurrent-cycles scenario generally shows better agreement with recently published data (Si et al., 2019; Witz et al., 2019) than single-process models (Figure 4—figure supplement 3). […] However, we note that part of this increase might also be caused by decreasing accuracy of detecting replication initiation during overlapping rounds of replication, which would artificially decrease correlations between replication and division.’

b. In the Discussion we added two paragraphs:

‘Based on the concurrent-cycles model we predict that inter-division and DNA replication/segregation processes are equally likely limiting cell division (p_H_ ≈ 0.5) in two different minimal growth media (Figure 4—figure supplement 3), and we previously reported the same balance (Micali et al., 2018b) for previous experiments at slow growth (Adiciptaningrum et al., 2015; Wallden et al., 2016). […] They demonstrate that temporal correlations between replication termination and z-ring constriction are high at slow growth, which supports a limiting role of DNA replication for cell division, but correlations decrease at fast growth, which then requires a different process to control cell division, in qualitative agreement with the concurrent-cycles model (Figure 4—figure supplement 3).’

and later

‘A qualitatively different behavior at slow growth was recently suggested in the already mentioned work by Tiruvadi-Krishnan et al., 2021. […] In the future, it will thus be interesting to re-investigate the balance between two different processes by implementing a variant of the concurrent-cycles model that considers an 'and' gate between replication termination and z-ring constriction.’

Reviewer #2 (Recommendations for the authors):Overall, this paper is well and clearly written.Figure 2.A. Are the two cells outlined below the same as the ones in the image series above? Does not seem so. There are too many "time points" in the below cartoon vs the above images and the positions of the spots don't correspond. This is hard to understand.

We previously showed different cells in top and bottom parts of panel A. We have now corrected this mistake.

B. It seems that cell lengths are greater in liquid culture than in mother machine at all A22 concentrations; the same might or might not be true of width. Why? Does this matter? Is it completely clear that the cells are in steady state after 6 hours (see discussion in Zheng et al., 2020)?

In the mother machine, the cells are sometimes a bit tilted, because their width is smaller than the channel width. This could result in a small error in segmentation and therefore an underestimation of cell length for cells cultured in the mother machine.

C. (i) Can the authors explain why, in the unperturbed case, the DnaN signal seems to "split" into two parts and then become single again? This does not happen in the A22 samples. Does this matter?

This is a very interesting observation. Upon qualitative inspection of single-cell traces treated with A22=0.25ug/ml we observed the splitting of foci is visible when only one chromosome is replicated in the cell, meaning when the number of oris goes from one to two, followed by cell division (left panel in Author response image 1), while replicases seem to stay closer together if replication happens across division events (right panel).

Since we observed the behavior within the same condition, without any qualitative differences between short lineages showing one or the other behavior, we do not think that this matters for the change of correlations observed. In response to the referee question regarding Figure 3, we now also separated point clouds of added lengths versus length at initiation into cells that initiation in the mother or in the daughter, respectively. We did not observe any significant difference in the slopes of the separate point clouds (now also added to a new Figure 3—figure supplement 2).For a more detailed account and the text added to the manuscript regarding this point see our answer to point Figure 3 (ii) below.

(ii) Also, it is noted that the times defined by these images do not "strictly agree" with average replication/termination times…Please explain? Are we supposed to see these as illustrating the general pattern of the change in replication initiation time, vs division, and the fact that C period is the same in all conditions, without worrying about the exact length of the C-period? Or…how are the average replication/termination times actually determined (vs these images)?

We understand that this discrepancy can cause confusion. First, cell-cycle sub periods shown in Figure 2C are obtained from single cells and subsequent averaging according to the rules specified in the methods. The peaks in the heat maps reflect both the temporal and spatial control of replicases. The dashed lines are meant as a guide to the eye. We have now highlighted this in the text, writing:

‘Vertical lines that indicate the beginning or end of peaks in Figure 2D are guides to the eye and should not be interpreted as average times of initiation or termination.’

We also realize that DnaN is stays bound to DNA about 5 min after termination [Moolman et al. Nat Communications 2014], which additionally shortens the period between termination peak and division (and increases the period between initiation and termination peaks), and we added this information to the text, writing:

‘Since DnaN stays bound to DNA for about 5 min after replication termination (Moolman et al., 2014), we likely overestimate the average C period and underestimate the D period by this amount. However, this absolute change of period durations does not affect our investigations of cell-cycle regulation, which are based on the combined C+D period.’

(iii) Also – it might be useful to indicate the "C" and "D" periods explicitly, eg in the unperturbed case? C period is time between red and yellow dotted lines and D period is time between yellow dotted line and next cell division, right? But in the A22 case, C period spans a division and D-period is again between yellow line and next division… This is obvious to people used to C+D but is something to keep in mind in terms of the biology.

As already written in response to the previous point the distances between lines must not be interpreted as average C or D periods, respectively. We thus decided not to indicate the periods in the heat maps.

Figure 3.– Legend has typo: B,D,F: Slopes of the added sizes corresponding to A, C, D should be "E", respectively

This is now corrected.

– In Panel E(i) There is an assumption involved in plotting the "average length added PER ORIGIN in C+D" versus the "initiation length PER ORIGIN". (i.e. in normalizing "per origin").The rationale for this approach should be made explicit. It is presumably derived from the Witz-type analysis in (C) which considers that everything scales with the number of origins. Perhaps this is in the SI/Methods, but it needs to be explained in the text.

We agree with the referee and have now included an additional sentence to motivate this assumption. Specifically, we now write:

‘Ho and Amir, 2015, previously demonstrated that the average size per origin and average added size per origin are equal to one another during steady-state growth. The scaling of average cell size at initiation with the number of replication origins initially deduced by Donachie,1968, and later confirmed for different growth rates

(Wallden et al., 2016) and for different cell widths (Zheng et al., 2020) is therefore also a strong motivation to consider the added size per origin (rather than the non-normalized added size) in our and previous single-cell studies (Si et al., 2019; Witz et al., 2019).’

(ii) Does it matter that in A22-treated cells, the C-period is spread over two division cycles? Probably not, since the C period is constant in all conditions, but this might be mentioned?

To test whether it matters for the correlations whether or not the C period is spread over two division cycles, we separated point clouds into those that initiate in the mother or in the daughter, respectively. This is possible, because untreated cells or cells treated with A22 (0.25ug/ml) sometimes initiate in the mother and sometimes in the daughter. If we separate the point clouds of added length during C+D period or added length during I period as a function of initiation size we observe that the two point clouds show approximately the same correlations. Here are example clouds for untreated and treated cells (now also added to a new Figure 3—figure supplement 2).

In the main text we now write:

‘With increasing average D period, replication is increasingly likely to happen in the mother cell (Figure \ 2D). […] We did not observe a separation of point clouds nor differences between their slopes, suggesting that the spread of the C period over a division event does not affect correlations between initiation and division or between subsequent initiation events.’

Figure 4.Panel AB. This is confusing (at least to this reviewer). In Panel A, the cartoon and legend say: "Two independent inter-division and timer-like replication/segregation processes with control parameters ζH and ζCD¨ = 1, respectively, must be completed before division occurs. The adder-like inter-initiation processes with control parameter ζI = 0 determines size at initiation." But in Panel B: "Model-fitting to experimental data reveals the probability pH of the inter-division process to control cell division as a function of increasing D period (with increasing A22 concentration), assuming constant control parameters ζH = 0, ζC+D¨ = 0, ζI = 0."Why does the model in (A) use different values for ζH and ζCD¨ than those used for the model fitting?

We thank the reviewer for spotting these typos. In both panels and in general for the model presented in the main text the interdivision (H) and the interinitiation (I) processes are assumed adders (ζ_H_ = 0 and ζ_I_ = 0). The minimum time for replication and segregation (the CD’ period) is considered a timer. We now updated the caption of Figure 4 to correct.

Reviewer #3 (Recommendations for the authors):1. It has been reported that MreB affected chromosome segregation (see Ram Madabhushi and Kenneth J. Marians, Molecular Cell, 2009, and Thomas Kruse et al., EMBO J, 2003 for details). The authors need to put this issue into discussion and/or analysis.

We added this information. See the response to the reviewer’s previous comment.

2. In the model, if QC+D' keeps constant, it is natural to conclude that 'inter-division process' would become more limiting (i.e., pH would increase), in the case of increased C+D periods. Since C+D' is unmeasurable, more discussion are need to explain the contribution of the experimental data to the conclusion of the model.

We cannot access Q_CD_’, experimentally, and Q_CD_’ is a necessary parameter of the model. We have underlined this limitation more clearly in the text. What we know however is that the loss of correlation between division size and initiation size under A22 treatment counters the hypothesis that D’ would increase as much as making the chromosome interfere with cell division in these perturbed conditions. In other words, if we were to make a model where D’ increases with A22 concentration, we would run into the problem of consistency with the CD period size correlation patterns, because if D’ increases replication-segregation becomes increasingly more limiting for division. We added the text already copy/pasted above.

3. In Figure 3F, it would be helpful to show a line indicating 'always depends on DNA replication'.

We added this information to Figure 3F and Figure 4.

4. Line 33, should be Figure 1BC.

This is now corrected.

5. Line 109-111, YPet-DnaN is used to measure periods of DNA replication. Since DnaN is reported to be associated with DNA for some time after replication is done (M. Charl Moolman et al. Nature Communications, 2014), one may overestimate the C-period in this manner.

Based on previous results by Moolman et al. Nat Communications 2014, which demonstrates that DNA-bound β-clamps are unloaded in about 5 minutes after replication termination, we overestimate the C period by this amount. We now write:

‘Since DnaN stays bound to DNA for about 5 min after replication termination (Moolman et al., 2014), we likely overestimate the average C period and underestimate the D period by this amount. However, this absolute change of period durations does not affect our investigations of cell-cycle regulation, which are based on the combined C+D period.’

6. There are three control parameters (ζ_H_, ζ_I_ and ζ_CD′_) in the model. The first two were assigned to 0 with explanation in Line 200-210. It needs more explanation of ζ_CD′_ in the main text.

ζ_CD’_ is assumed to be a timer, i.e. we assume that these processes are size-uncoupled and that all the correlations in the CD period come from the (D-D’) period. This might not be the case, but our data do not allow us to test alternative hypotheses. We added additional explanations about this in the main text writing:

‘Furthermore, we assumed that replication segregation is a timer process (ζ_CD'_ = 1) that requires a minimum time to be completed, which is independent of size at the time of initiation, and does not vary in A22 perturbations.’

We also added more explanation to the caption of Figure 4.

7. In Figure 2B and 2C, standard deviations for each gray rectangle are missing.

To keep Figure 2 readable and because the goal of the panels is to show changes in averages, we added the coefficient of variations for the different panels of Figure 2C to Figure 2—figure supplement 3. Interestingly, interdivision time, I and C periods Coefficient of variation are invariant when cell width is modified. The coefficient of variation of D period is slightly reduced when cell width is increased, largely due to an increase of the average D period. Panel added in Figure 2—figure supplement 4:

We also refer to this figure in the manuscript text, writing:

‘Cell-to-cell fluctuations in the duration of sub-periods remain constant (I, C, and interdivision periods) or decrease mildly (D period) (Figure 2—figure supplement 4B).'

[Editors' note: further revisions were suggested prior to acceptance, as described below.]

The manuscript is essentially ready for publication. Please address the two remaining comments from referee 3 as you see fit, for example by editing the two figures as suggested by the referee.Reviewer #3 (Recommendations for the authors):The revision makes the manuscript much more comprehensible. I have two remaining comments:1. Since most cell size-related models emphasize steady-state, providing further evidence that the single-cell datac here were acquired during steady-state is important, as also commented by Reviewer 1. Similar to Figure 2-Supplement 2, I would recommend plotting the 'inter-division time' vs. 'Time of birth' as well, because growth rate is likely to be constant with continuous medium supply but 'inter-division time' may not, whereas steady-state requires a similar rate for both cell growth and division. Furthermore, showing these two plots for the highest A22 dosage (1 ug/mL) data or even all dosages used in the work would be a very strong evidence that the cells were in steady-state.

We agree with the referee and have now plotted the interdivision time and growth rate as a function of birth time for all four conditions in the updated Figure 2–Supplement 3. During the full measurement window of >6h, none of these quantities nor width or length show variations of >15% as already previously indicated in the Methods. We added the information about the interdivision time to the Methods, writing:

‘We made sure that cells were growing at steady state in terms of growth rate/interdivision time/length/width for at least six hours.’

2. 'Initiation mass' (or initiation volume) is commonly used to refer cell size upon initiation. With altered cell width in this work, cell length is no longer a good proxy for cell mass, especially in important figures like Figure 3E. therefore, I strongly recommend to use cell volume in both axis in Figure 3E (or entire Figure 3) to precisely convey the message that the added mass becomes negatively correlated with initiation mass. The dots in Figure 3E (right panel) may also be less clustered and the trend line could be more significant, if volume is used there.

We agree with the referee that it would be interesting study correlations in cell volume. Even better would be to study correlations in cell mass. Whether volume or surface area are better proxies for single-cell mass is still to be studied (see our recent publication by Oldewurtel et al. PNAS 2021, which shows that the ratio of surface to mass remains constant during the cell cycle while dry-mass density varies systematically with length). However, in our data, both surfacearea and volume calculations are subject to substantial measurement noise in width, so that (within conditions) the best available proxy for both volume and surface area is actually length. Specifically, we and others measured relative width variations to be about 10% in the mother machine, but physical cell-to-cell variations are likely about 5% – see our measurements on agarose pads in Figure 2—figure supplement 2 and our recent publication by Oldewurtel et al. PNAS 2021. Hence, while absolute uncertainty in width and length measurements are likely very similar, measurement noise in width leads to much higher uncertainty in volume. In quantitative terms, the same absolute uncertainty in width as in length leads to ~6-fold larger relative uncertainty in volume than observed in length alone. Furthermore, measured cell-to-cell fluctuations in width do not increase with increasing drug concentrations, suggesting that the observed decrease of correlations between initiation size and division size (with increasing A22 concentration) is not a consequence of width fluctuations. Consistently, correlations between birth and division size, and between subsequent initiation sizes remain nearly constant, respectively. We thus decided to use length variations as a proxy for cell-size variations.

Regarding the changes *across* conditions, we note that our conclusions on size correlations are based on size fluctuations around their respective means, and thus they are not affected to meanwidth changes across conditions (neither our qualitative conclusions from experiments nor our theoretical model are based on average initiation or division mass).

We have now laid out these arguments in a new section in the Materials and Methods part titled ‘The use of length fluctuations as a proxy for size fluctuations’.